# Energy Metabolism and Ketogenic Diets: What about the Skeletal Health? A Narrative Review and a Prospective Vision for Planning Clinical Trials on this Issue

**DOI:** 10.3390/ijms22010435

**Published:** 2021-01-04

**Authors:** Daniela Merlotti, Roberta Cosso, Cristina Eller-Vainicher, Fabio Vescini, Iacopo Chiodini, Luigi Gennari, Alberto Falchetti

**Affiliations:** 1Department of Medicine, Surgery and Neurosciences, University of Siena, 53100 Siena, Italy; dmerlotti@yahoo.it (D.M.); luigi.gennari@unisi.it (L.G.); 2Istituto Auxologico Italiano “Scientific Institute for Hospitalisation and Care”, 20100 Milano, Italy; robcos63@libero.it (R.C.); i.chiodini@auxologico.it (I.C.); 3Unit of Endocrinology, Fondazione IRCCS Cà Granda-Ospedale Maggiore Policlinico Milano, 20122 Milano, Italy; eller.vainicher@gmail.com; 4Endocrinology and Metabolism Unit, University-Hospital S. Maria della Misericordia of Udine, 33100 Udine, Italy; fabio.vescini@asufc.sanita.fvg.it; 5Department of Medical Biotechnology and Translational Medicine, University of Milan, 20122 Milano, Italy

**Keywords:** ketone bodies, ketogenic diets, energy metabolism, bone health, bone mass, bone turnover markers, DXA scans, fragility fractures

## Abstract

The existence of a common mesenchymal cell progenitor shared by bone, skeletal muscle, and adipocytes cell progenitors, makes the role of the skeleton in energy metabolism no longer surprising. Thus, bone fragility could also be seen as a consequence of a “poor” quality in nutrition. Ketogenic diet was originally proven to be effective in epilepsy, and long-term follow-up studies on epileptic children undergoing a ketogenic diet reported an increased incidence of bone fractures and decreased bone mineral density. However, the causes of such negative impacts on bone health have to be better defined. In these subjects, the concomitant use of antiepileptic drugs and the reduced mobilization may partly explain the negative effects on bone health, but little is known about the effects of diet itself, and/or generic alterations in vitamin D and/or impaired growth factor production. Despite these remarks, clinical studies were adequately designed to investigate bone health are scarce and bone health related aspects are not included among the various metabolic pathologies positively influenced by ketogenic diets. Here, we provide not only a narrative review on this issue, but also practical advice to design and implement clinical studies on ketogenic nutritional regimens and bone health outcomes. Perspectives on ketogenic regimens, microbiota, microRNAs, and bone health are also included.

## 1. Introduction

During the evolution, the mankind has gone through alternating periods of famine/abundances, determined by the seasons and environmental condition changes, consequently inducing switching metabolism efficiency. Of course, the capacity of adaptation and adjustment to these changes has helped us to survive as a species. Currently, in developed countries, radical diet fluctuations are extremely rare, and in this sense, the human metabolism is largely “unchallenged”, but whether or not this represents a favorable aspect is hard to assess. To date, an important aspect to be considered is not only the quantity of ingested food, but also its quality and nutrients combination. Indeed, obesity and type 2 diabetes mellitus (T2DM) are approaching epidemic proportions, which is also a consequence of a “poor” quality in nutrition. Through the metabolism, our body draws the necessary energy from food, during both rest and physical activity. The nutrients in food, summarized for convenience in carbohydrates (CHO), proteins and fats, are broken down during digestion into smaller molecules: glucose, amino acids, and fatty acids, respectively. CHO are the main source of energy for the human body. Interconnections between glycolysis, tricarboxylic acids cycle (TCA) and ketogenesis exist, as summarized in Figure 1. Each oxidized glucose molecule degraded within the glycolysis pathway may provide an effective yield of 30–32 molecules of the most useful energy molecule known as adenosine triphosphate (ATP). Additionally, proteins and lipids degradation contribute to ATP formation. The term “ketogenic”, in general, indicates the capacity to stimulate the production of ketone bodies (KBs) by various cells in our body, when a supplementation of less than 100 g of glucose is provided in the diet.The most recent knowledge of bone pathophysiology shows evidence of the role of the skeleton in energy metabolism as well, in particular in glucose homeostasis. However, both basic and clinical studies on the possible relationship between ketogenic nutritional regimens and skeletal health are still needed, which is also in consideration of the vast “audience” of patients for whom ketogenic diets (KDs) can be of great help, other than epileptic subjects resistant to specific therapies.

In this review, we report the available data regarding bone cells energy metabolism physiology, bone pathophysiology, cell-/animal-model investigations and human clinical studies with relation to KDs and bone health. Moreover, based on the author’s experiences, practical and theoretical advice for the design and implementation of clinical studies on ketogenic nutritional regimens and bone health outcomes are provided. Finally, the perspectives on the possible relationship between inflammation, KD regimens, microbiota, microRNAs, and bone health have also been explored.

According to PRISMA guidelines, PubMed and MEDLINE were searched from March 1990 to November 2020 to identify published articles about KDs and bone health, bone mineral density (BMD) and content (BMC), bone metabolism, bone turnover markers (BTMs), fragility fractures, inflammation, microbiome, and microRNAs. The clinical studies that analyzed the KDs impact on bone health and fragility were evaluated. Only English-language publications were included.

## 2. Ketogenic Diets

KDs are normal-high-fat and low-CHO diets that respect an adequate protein regimen, and this proved to be effective in epilepsy in the last 100 years, with potential mechanisms including the direct anticonvulsant effect and the reduced neuronal excitability induced by KBs. Interestingly, it has been shown that KBs produce more energy than glucose due to the metabolic effects of ketosis. In particular, the high chemical potential of 3-β-hydroxybutyrate (3βOHB), the major KB in human metabolism, leads to an increase in the Gibbs free energy (∆G0), or the thermodynamic potential that is minimized when a system reaches chemical equilibrium at constant pressure and temperature [1], from the hydrolysis of ATP molecules. The blood sugar levels, although reduced, remain within the physiological range since glucose in blood comes from two sources: (a) the glucogenic amino acids [alanine, arginine, asparagine, aspartic acid, cysteine, glutamic acid, glutamine, glycine, histidine, methionine, proline, serine, and valine (Figure 1)], and (b) the glycerol released from triglycerides of lysine residue (lysine is a ketogenic amino acid (Figure 1)).

In long-term follow-up studies, approximately 20% of children treated with KD (≥6 years) experienced an increased incidence of bone fractures [2]. Bone fractures and decreased bone mineral density (BMD) are of concern for children maintained on KD therapy (KDT) and chronic antiepileptic drugs (AED) [2,3,4,5,6]. The causes of such negative effects on bone health may be due to the KD itself, to the high “acid load” milieu via the KBs, to alterations in vitamin D levels, to decreased levels of growth factors, together with both the concomitant use of AEDs, and the reduced/absent mobilization observed in a large proportion of patients with intractable epilepsies [5,6].

### 2.1. Forms of KDs

Schematically, four major forms of KDs can be described: (1) the “classic” KD; (2) the modified Atkins diet (MAD); (3) the medium chain triglyceride diet (MCT); and (4) the low glycemic index diet (LGID). These regimens are mainly differentiating in the lipids to the protein/CHO ratio [7]. More recently, Very Low Calories Ketogenic Diet (VLCKD) has been developed as a nutritional intervention mimicking fasting through a marked restriction of daily intake of CHO, usually <30 g/day (≃13% of total energy intake), with a relative increase in the proportions of fats (≃44%) and proteins (≃43%) and a total daily energy intake <800 Kcal. Importantly, VLCKD should not be considered as a hyper-protein diet since its daily protein intake is around 1.2–1.5 g/kg of ideal body weight [8]. Weight loss in obese subjects is able to reduce the prevalence of related complications, in terms of type 2 diabetes mellitus (T2DM) prevention and severity, hypertension, dyslipidemia, sleep apnea, fatty liver disease, osteoarthritis, stress incontinence, gastroesophageal reflux, and polycystic ovary syndrome [8]. Consequently, several guidelines, including the Italian ones, have listed specific metabolic diseases in which VLCKD is indicated in the clinical management of patients. Unfortunately, due to the lack of adequate studies, bone and mineral metabolism diseases are still not considered as possibly influenced by KD [8].

### 2.2. Are KBs the Only Main Source of Energy in Fast and/or Low Carbohydrate Periods?

During a prolonged fasting, gluconeogenesis removes intermediate products from the Krebs cycle or TCA (Figure 1) and directs acetyl-coenzyme A (CoA) towards KBs, which may also have a role in regulating food intake. The main KBs in humans are represented by 3βOHB, acetone and acetoacetic acid (Figure 1) generated by the liver from fatty acids [9] during periods of low food intake (fasting), carbohydrate restrictive diets, starvation, and prolonged intense physical exercise [10]. They represent ancient fuel substrates, evolutionarily preserved, and a fundamental energy source for heart, skeletal muscle, kidney, and brain. It has been suggested that KBs, in particular 3βOHB, preserve muscle protein by the occurrence of systemic inflammation, also participating to the metabolic defense against insulin-induced hypoglycemia [11]. Moreover, 3βOHD is not only the substrate for ATP production (Figure 2), but binding to specific hydroxyl-carboxylic acid receptors (HCAR), inhibits histone deacetylase (HDAC) enzymes, free fatty acid receptors (FFAR), and the NOD-, LRR- and pyrin domain-containing protein 3 (NLRP3) inflammasome, thus acting as a signaling molecule, promoting the transcription of genes for oxidative stress resistance factors [12] and ultimately influencing, at an epigenetic level, bone cell physiology [11] (Figure 3).

## 3. Skeleton and Energy Metabolism: A Complex “Multiplayers Ping Pong Match Model”

Bone cell progenitors, skeletal muscle cells and adipocytes share a common mesenchymal stem cell progenitor, and the bone cell progenitors can be redirected to become fat cells. These mechanisms evolved to acquire and store fuel, and so it is no longer surprising that the skeleton could also play a role in energy metabolism. In the early tetrapods, a vertebrate superclass with four limbs prevalently adapted to life in a subaerial environment; the evolution of a large appendicular skeleton, powered by robust skeletal muscles, was a successful strategy for walking on the ground, and this general concept can also be extended to mankind. This strategy also justified the need for a “new” skeleton as storage for the calcium contemporary, providing hormonal/molecular mechanisms for the “fast” calcium-phosphate removal from the bone. The development of the parathyroid glands, producing the parathyroid hormone (PTH), was equally necessary. According to its needs, our body may orchestrate different appropriate energy strategies to determine or imbalance with the loss of bone mass (and osteoporosis) or muscle/fat disorders [13]. Over the past 25 years, some findings have suggested human skeleton to play a role in energy metabolism through “local” hormones, such as adipokines, Insulin/Insulin-like growth factor-1 (IGF-1), osteocalcin (OC)/undercarboxylated osteocalcin (UcOC) pathways [13,14,15], and bone morphogenic proteins (BMPs) [16], in cooperation with organs involved in metabolic control (e.g., skeletal muscles, small intestine, and endocrine pancreas) [13]. This complex molecular multidirectional network may be fundamental in maintaining energy homeostasis by controlling and coordinating both “fuel” uptake and energy expenditure, probably within an equally complex hierarchical order in which the central nervous system and peripheral energy centers, sensing and regulating the energy needs, cooperate.

### 3.1. Bone Remodeling Needs of Boosting Energy

Bone remodeling is a continuous skeletal metabolic structural adaptation to stress forces, fundamental to always ensure a “dynamic” bone structure to adequately satisfy the different biomechanical needs throughout life. To be effective, bone remodeling must have an extremely coordinated regulation, in time and space, of bone resorption and new bone formation phases [17,18], by systemic and local release of cytokines and growth factors [13,14,15,16]. Osteoblasts (OBLs), lining cells, osteoclasts (OCLs), and osteocytes (Ocs) are ultra-organized into the so called “bone remodeling units” (BRUs) with a highly regulated coupling of the specific bone cell functions [19]. There are approximately two million BRUs working in each individual at any given time. The first step of the remodeling cycle consists of bone resorption by OCLs, the “pick hammers”, which release protons and specific proteases (an energy consuming process), thus creating a highly acid environment, which is an essential prerequisite for bone resorption. Later, stromal-derived mesenchymal cells, recruited at the newly excavated site, differentiate into OBLs, the “bricklayers”, and are able to affix and mineralize the “new” freshly formed bone [20] by the production and secretion of specific proteins, particularly type I collagen. All these biological processes require adequate and appropriate energy expenditure. Once bone formation is completed, OBLs give rise to Ocs [21] that act as mechanical sensors [22] and secrete specific osteokines, such as sclerostin. Sclerostin binds to the low-density lipoprotein receptor-related proteins 5 and 6 (LRP5/6) receptors on OBLs cell surface and inhibits the Wnt signaling pathway determining anti-anabolic effects on bone formation [23]. In humans, the complete remodeling process takes approximately 100 days and the entire skeleton is remodeled every 10 years. Consequently, a high energy cost must be incurred [24].

### 3.2. Insulin, Osteocalcin, Osteoprotegerin, Receptor Activator of Nuclear Factor Kappa-B Ligand: A Complex, Interconnected, Network

At the skeletal level, insulin signaling stimulates either the OC expression or the OBLs differentiation, by inhibiting Twist2, an inhibitor of the osteoblastogenic factor Runt-related transcription factor 2 (RUNX2), also known as the core-binding factor subunit alpha-1 (CBF-alpha-1), encoded by the *RUNX2* gene [25]. The expression of Forkhead box protein O1 (Foxo1), a transcription factor (TF) encoded by *FOXO1* gene [26], is higher in skeletal tissues and both Foxo1 activity and the expression increases in mouse mesenchymal cells under osteogenic stimulants. The Foxo1 silencing blocks the expression of Runx2 as well as other osteogenic markers, such as alkaline phosphatase (ALP) and OC, thus reducing the calcification process, even in the presence of strong osteogenic stimulants. Therefore, the main mechanism, through which Foxo1 affects mesenchymal cell differentiation into OBLs, occurs exactly through the regulation of Runx2 [27]. Moreover, Foxo1 plays important roles in the regulation of both gluconeogenesis and glycogenolysis by insulin signaling and it is also central to the decision for preadipocytes to commit to adipogenesis [28]. This TF is primarily regulated through phosphorylation on multiple residues, and its transcriptional activity is dependent on its phosphorylation state, resulting in the accumulation of carboxylate OC in the bone matrix. Conversely, insulin activates OCLs and accelerate bone turnover via the increasing of the ratio of the osteoprotegerin/receptor activator of nuclear factor kappa-B ligand (OPG/RANKL). Activation of OCLs determines decarboxylation of the bone matrix-embedded OC, which, in turn, reduce OC affinity to the matrix itself with consequent UcOC release into the circulation. The circulating UcOC stimulates the expression of insulin in pancreas and adiponectin in adipose tissue, both expressing the cytosolic OC specific receptor, GPRC6A, also expressed in testis tissue [29]. Table 1 describes the main hypothesized endocrine UcOC functions.

#### Genetic Data on UcOC in Glucose Metabolism Regulation

Interestingly, genetic data support the fact that, through UcOC, the skeleton contributes to regulating glucose metabolism: (i) heterozygous dominant negative mutation of *GPRC6A* gene in two patients has been associated with peripheral testicular failure, glucose intolerance, insulin resistance, and increased body mass index (BMI) [30]; (ii) a significant association between *BGLAP* (OC gene) genetic variants and BMI in healthy subjects has been reported to be most likely associated with body mass as composite phenotype and less likely associated with adipose tissue itself, even though not only the *BGLAP* gene variants may cause this association [31]; (iii) *R94Q* Single Nucleotide Polymorphism (SNP) of *BGLAP* exon 4, near to one γ-carboxylation site of OC, associates with insulin sensitivity and glucose disposal in Afro-Americans [32]. However, DNA polymorphic variants of *BGLAP* have been excluded as a major risk factor for T2DM in Caucasians [24]; and (iv) patients with autosomal dominant osteopetrosis, due to an OCL activity deficit, exhibited reduced levels of UcOC together with hypoinsulinemia [33]. Interestingly, if exposed to chronic high glucose levels, the bone marrow stromal stem cells (BMSCs) show an enhanced adipogenic activity due to both the enhancement of the peroxisome proliferator-activated receptor gamma (PPARγ)-dependent pathway, and the of cyclin D3 expression increase [34], which, in turn, is associated with a decreased Runx2 [35], ALP [36], and OC expression in OBLs.

In summary, the skeleton can achieve its role in maintaining glucose homeostasis through the interaction with different systems/organs, such as endocrine glucose-regulation of the pancreas, liver, white adipose and skeletal muscle tissue, especially when a general increase in fuel requirements is necessary.

## 4. The Role of Wingless-Type Mouse Mammary Virus Integration Site (Wnt) Pathway in Bone Pathophysiology and Energy Metabolism

The Wnt signaling enters into many biological processes, including embryogenesis, postnatal development, and tissue homeostasis in adults [37]. More recently, the Wnt signaling also revealed a regulatory role in bone formation and healing [38], so that proteins as sclerostin, LRP5/6, and Dickkopf-related protein 1 (Dkk-1), which are crucial actors of this signaling pathway, have been the targets of randomized clinical trials (RCTs) for the development of anti-fracture therapies [39]. Studies in animal models confirmed a reduced mineralization rate and a reduced trabecular bone volume in T2DM subjects, probably due to the decreased *RUNX2* gene expression and reduction of OC, OPG, BMP-2, and ALP expression [18,40,41,42,43,44]. Specifically, BMP signaling was demonstrated to interact with several signaling pathways such as the mammalian target of rapamycin (mTOR), activated by amino acids, glucose, insulin, hypoxia-inducible factor (HIF), Wnt, and self-degradative process autophagy to coordinate both energy metabolism and bone homeostasis [44].

According to such findings, it can be suggested that some forms of osteoporosis may occur as a consequence of several impairing energy metabolism disorders, as T2DM and anorexia nervosa. Changes in bone cell energy metabolism have significant roles in regulating the differentiation and function of bone cells.

### 4.1. Bone Cells-Specific Bioenergetic Program

Differentiation stage-specific bioenergy programs exist in bone cells [44], which are also requirements for specific different energy programs at distinct stages of maturation/function of the OBLs and OCLs. In fact, proliferative BMSCs and cells already at an osteogenic differentiation stage essentially use glycolysis as an energy source, even if the underlying mechanisms of glucose metabolism involvement at this “early” stage of cell maturation are still to be clearly defined [45,46]. In rats, the essential amino acids effect on the proliferation and differentiation of osteoprogenitors depends on IGF-1 signaling [47,48]. After osteogenic induction, OBLs switch to use both glycolysis (including aerobic glycolysis) and the TCA (Figure 1) [48,49,50,51,52,53].

#### 4.1.1. “Fuel” Selection according to Bone Cells Differentiation Steps

The glucose metabolism has several “controllers” during bone cells differentiation, such as Glucose transporter 1 (GLUT1)-RUNX2 [54], hypoxia-inducible factor 1α (HIF-1α), a TF sensitive to the decreased availability of oxygen in the cellular environment, or hypoxia [55], PTH, IGF-1 [56], or Wnt-mTORC2 signaling [57]. Interestingly, either pathological or environmental hypoxic conditions appear to influence bone health, since bone cells are distinctly responsive to hypoxic stimuli, even though it is still unknown whether or not they act in a negative or positive way. It has been suggested that hypoxia may induce an osteogenic-angiogenic response, but also stimulate excessive OCL activity. Moreover, several hypoxia-associated factors may influence bone metabolism by determining changes in energy metabolism and increasing the generation of reactive oxygen species (ROS), mechanisms that may impair the physiological acid–base balance [58]. Practically, there is a metabolic reprogramming during OBL differentiation so that, while the aerobic glycolysis (one molecule of glucose is split into two pyruvate molecules) increases during the OBL differentiation process (Table 2), the mitochondrial respiration and energy production progressively decrease, reaching their minimum in the mature OBL [59]. In the presence of low O_2_ concentrations, pyruvate is converted into lactate, which, in turn, determines the increase of (a) formation of ALP positive (ALP^+^) cells; (b) ALP activity; and (c) expression of osteoblastic differentiation markers. Lactate, together with citrate, would facilitate bone turnover and contribute to the global solubility of mineral ions in the extracellular bone matrix [60]. Table 2 depicts the main biochemical process underlying metabolic reprogramming during osteoblast differentiation.

Therefore, the TCA cycle represents the final common pathway for the oxidation of CHOs, lipids, and proteins because glucose, fatty acids, and most amino acids are metabolized to acetyl-CoA or intermediates of the cycle.

The Wnt-related “governance” of fuel selection by OBLs mostly depends on their differentiation status. Immature OBLs increased the use of glucose and glutamine (Gln), whereas the mature cells switch to increase the use of fatty acids. Differentiated OBLs, engaged in matrix synthesis and mineralization, exhibit abundant mitochondria for the high energy demands of protein synthesis. This cell differentiation phase associates with β-oxidation of fatty acids, producing approximately more than 100 ATP molecules per palmitate molecule, in response to Wnt anabolic stimulation. Wnt activation inhibits the entry of glucose into the TCA cycle, regulating the availability of substrates for histone acetyltransferases, epigenetic enzymes, which are able to install acetyl groups onto lysine residues of cellular proteins such as histones and others [61].

#### 4.1.2. Glutamine and Wnt-Dependent Bone Metabolism

Glutamine is a polar amino acid, and its enantiomer L form represents one of the 20 ordinary amino acids, essential for the metabolism of the nervous system being also a basal intermediate of liver and kidney functions. In a fasted bird model, the rate of Gln release from skeletal muscles increases when elevated concentrations of KBs are present, which are due to the increased availability of glutamate for Gln synthesis [62]. Moreover, in vitro studies on skeletal muscle cells from fasted mammalian and avian species, revealed that KBs inhibit both protein synthesis and protein degradation in skeletal muscles, probably representing an important adaptation, survival, a mechanism in prolonged fasting, or in general, in catabolic states. However, since Gln shares transporters with other amino acids, its increased intake or cell/tissue availability may alter the tissues distribution of amino acids and their absorption in the intestine and kidneys [63]. It is currently unknown whether these mechanisms also occur at the bone cells level, as a whole or with a cell-specificity. Furthermore, it is not conceptually correct to translate the information obtained in skeletal muscle cells directly to the bone cells. Recent studies indicate Gln and fatty acids as potential energy sources used by OBLs [64,65], confirming that, when glucose availability is scarce, most tissues use fatty acids as an energy source, or convert other substances into sugars. In a hyperactive Wnt signaling mouse model of human osteosclerosis, the inhibition of Gln, together with the suppression of the excessive bone formation, indicates that Gln metabolism may mediate the in vivo “Wnt-induced” bone anabolism. Consequently, Gln could act as both an energy source and a protein-translation rheostat, responsive to Wnt, and the handling of Gln signaling pathway could represent a future pharmacological target to correct the deranging protein anabolism in human diseases [66]. This mechanism is likely to be very sophisticated, and in immature OBL, the Wnt-mediated activation of mTORC1 increases the abundance of glutaminase, the first enzyme in glutaminolysis, whereas the mTORC2 complex activation regulates the abundance of proteins involved in glycolysis, as an epigenetic mechanism [67].

#### 4.1.3. Inhibiting the Inhibitor of the Wnt-Dependent Differentiation and Maturation of OBLs: Improvements of the Insulin Sensitivity in an Animal Model

Sclerostin, encoded by the *SOST* gene, is expressed by Ocs and articular chondrocytes and represents an endogenous inhibitor of Wnt signaling, and consequently, of both OBL differentiation and maturation. Interestingly, sclerostin overproduction in adipocyte cells increases fat mass by regulating catabolic and anabolic metabolism. In a Sost^−/−^ knock-out model of diabetic mice, an increased insulin sensitivity and a reduction of fat tissue accumulation, as compared to the wild-type, has been described. Sost^-/-^ mice and mice treated with anti-sclerostin antibodies were resistant to metabolic disorders induced by an obesogenic diet. The white adipose tissue mass reduction and protective effects against the high fat content diet might be in relation to an increased oxidation and reduced de novo synthesis of fatty acids in the adipocytes due to the increase of Wnt/β-catenin signaling [68]. However, whether or not this mechanism represents the result of the real effects of sclerostin on Wnt signaling and white adipose tissue metabolism is still unclear. It should be considered that sclerostin expression by Ocs is inhibited by either PTH or mechanical stress. Interestingly, glycolysis is also predominant in fully differentiated OBLs, but the underlying molecular mechanisms are still unknown. The differentiation and function of OCLs also need energy boosts and glucose represents an important energy source for them. Unlike what occurs in OBL, in the progression of differentiation in OCLs, the TCA cycle dominates over glycolysis, although the latter seems to be critical for bone resorption. Glucose metabolism in OCLs depend on mitochondrial nicotinamide adenine dinucleotide (NAD)-dependent deacetylase sirtuin-3 (Sirt3)-5′ AMP-activated protein kinase (AMPK) [69,70], HIF-1α, the transcription factors family MYC, and p38-MAPK signaling [44]. What role amino acids and fatty acids may play in the energy metabolism in OCL differentiation and function is still unknown.

Finally, even if Ocs represent the most prevalent bone cells (~95%) in the adult skeleton, [71], data on their bioenergetics metabolism are virtually lacking, and which could be their preferential fuel sources and bioenergetic programs is still unknown. More details on this aspect are reported in the review by Yang et al. [44].

### 4.2. Osteoanabolic Signals Stimulate Glycolysis

It has been suggested that PTH stimulates aerobic glycolysis via IGF-1. The PTH-dependent «engage» of IGF-1 enhances the activation of mTORC2, which is involved in the skeleton growth and in the aerobic glycolysis induction [56]. Osteoanabolic PTH, whose receptor (PTHR) is expressed in the osteoblastic lineage, promotes glycolysis “in vitro”, and the suppression of the glucose uptake blows the “in vivo” PTH anabolic effect. Moreover, PTH increases both the glucose absorption rate and O_2_ consumption in OBL. However, PTH suppresses glucose entry and progression of glycolytic metabolites into the TCA cycle, but, currently, it is not clear if OBLs do not get the full energy benefit of the glucose consumed. Several questions are still open regarding all this, and in particular: (i) it is a large portion of the pyruvate generated by glycolysis converted and secreted as lactate and why carbon derived from glucose cannot be used in biosynthetic routes? Studies of early radio-tracing demonstrated the presence of ^14^C-glucose molecules in bone collagen [60,72]; (ii) could pyruvate and lactate act as «scavengers» to protect bone from cytotoxic oxidants? Moreover, it could not be excluded that PTH may play a role in other metabolic substrates. Indeed, PTH suppresses glucose oxidation in the TCA cycle, while simultaneously inducing a paradoxical increase in O_2_ consumption. A furthermore plausible explanation for this is that PTH promotes the oxidation of other substrates to generate ATP (see later in the text). However, it is still unknown whether PTH “orders” the OBLs to convey glucose metabolites toward amino acid synthesis. Understanding whether or not this mechanism may actually take place at the cellular level would have important repercussions in improving our knowledge of the pathophysiology of bone cells. In order to unravel the possible bone cellular molecular mechanisms, it would first be appropriate to investigate whether glucose homeostasis, more generally, really depends on the existence of a direct/indirect relationship between production/secretion of PTH and insulin, as described below.

#### Is the Existence of an Insulin/PTH Axis Conceivable?

In 2000, Clowes et al. suggested that the growth hormone (GH)/IGF-1 axis may be involved in the differential suppression of PTH following oral and intravenous (i.v.) glucose [73]. Lately, it has been reported that, in non-diabetic post-menopausal osteoporotic women with primary hyperparathyroidism, the administration of PTH (1–84) reduces basal blood glucose and this effect can be mediated by changes in serum OC levels rather than PTH-induced glucose uptake by peripheral tissues [74]. A small clinical study on eight healthy, normal glucose-tolerant, individuals, with a negative familial history of diabetes, and on nine patients with type 1 diabetes mellitus (T1DM), with C-peptide exhibiting a negative response to 5 g i.v. arginine test, demonstrated a more pronounced inhibition of PTH secretion in healthy individuals versus T1DM patients during OGTT. Covariance analysis showed that insulin levels correlated significantly and inversely with PTH levels, thus suggesting that insulin may act as an acute regulator of PTH secretion in humans via insulin binding to insulin receptors, IGF1R and IGF2R, expressed in human parathyroid cells [75], as already suggested by other authors [76].

### 4.3. What Do We Know about the Role of IGF-1 in Bone Health?

IGF-1 may have anabolic effects on bone, particularly during the acquisition of bone mass in adolescents and the maintenance of skeletal architecture in adults, parameters potentially affecting the risk of subsequent fracture [77,78]. Results from observational studies on the role of IGF-1 for bone health have been conflicting, some authors finding a reduced risk of both BMD loss and fracture in the presence of high serum IGF-1 levels [79,80,81,82], while others did not confirm these findings [83,84]. Two RCTs provided conflicting results concerning the skeletal effects of a one-year treatment with IGF-1. The former RCT revealed an attenuation of the proximal femur bone loss, together with a shorten hospital stay among patients with recent hip fracture [85], treated with IGF-1. The latter RCT, on 16 individuals treated with IGF-1 for 12 months, showed no effects on BMD [86]. Table 3 describes the potential relationship linking IGF-1 levels to 3βOHB levels in famine and IGF-1 levels to KDs.

### 4.4. Other Energetic Substrates in OBLs

Amino acids and fatty acids metabolism appear to be relatively less stringently controlled, the former by HIF-1α and activating transcription factor 4 (ATF4) [55,66,67], while the latter by Wnt-CD [64,66,91,92], Wnt-β-catenin [62,91,92,93,94], and GPR120, CD36 [94,95,96]. The cluster of differentiation antigens (CDs) are cell surface proteins that can act as receptors. The oxidation of fatty acids is sensitive to cyclic adenosine monophosphate (cAMP)/protein kinase A (PKA) signaling activated by PTH [97,98]. The β-oxidation of fatty acids is capable of producing KBs from food or storage lipids, and interestingly, it has been described that the pharmacological inhibition of β-oxidation alters the OBL differentiation in vitro [64].

In summary, glycolysis, gluconeogenesis, and oxidation of fatty acids regulate the availability of acetyl CoA, and therefore, the KBs formation.

## 5. How Could KDs and KBs Affect Bone Metabolism?

On the basis of all the evidence reported, it is not surprising that KDs may have an effect on bone. This relationship is also sustained by in vitro data. Indeed, in an in vitro model of OBL cultures, different ALP and mineralization activities have been shown to depend on the type of prevalent KB. Specifically, the OBL mineralization activity seems to be up-regulated by acetoacetate and down-regulated by 3βOHB [99]. Moreover, high-fat diet (HFD)-induced obesity determines the increases of the acetoacetyl-CoA synthetase (*AACS*) gene expression, involved in utilizing KBs for the fatty acid-synthesis during adipose tissue development [100]. A study on HFD-fed mice model, revealed that *AACS* was also expressed at both the backbone region and adult femur epiphysis, at variance with other embryo regions. Interestingly, HFD-fed mice exhibited higher AACS mRNA expression in the adult femur versus normal-diet fed mice. This mechanism was not observed in those mice whose obesity was induced by a high sucrose diet. Gene expression increases were also reported for 3-hydroxy-3-methylglutaryl coenzyme A reductase (*HMGCR*) and interleukin (IL)-6 genes, in HFD strand. Higher AACS mRNA expression was found in RAW 264, a differentiated OCL cells line, significantly up-regulated by IL-6 treatment that induce bone resorption. These findings may possibly be considered as a novel function of the KBs in bone metabolism, also indicating AACS and KBs as a sort of molecular bridging factors between obesity and osteoporosis. In a mouse model, the 3βOHB and its derivative methyl ester 3βOHB inhibited the development of osteoporosis in mice maintained under simulated microgravity, preserving both the bone microstructure and mechanical properties. These KBs inhibit the activation of the nuclear factor of activated T cells DNA-binding transcription complex 1 (NFATC1), a master transcription regulator of OCL differentiation [101], preventing the differentiation and maturation of pre-OCLs. Ultrastructure analysis by micro-CT in mice revealed that a dose of 100 mg/kg of KBs produced the most obvious effect on the prevention of osteoporosis by reducing calcium concentrations, as an indirect index of reduced OCL activity, and increasing the bone volume/total volume ratio (BV/TV), Trabecular Number (Tb.N), Trabecular Thickness (Tb.Th), and Trabecular Spacing (Tb.Sp) indexes at the bone specimen levels [102].

### 5.1. Animal Studies on KDs and Bone Metabolism

Low BMD, abnormal cancellous and cortical bone mass have been described in mice under KD treatment [103]. Quite recently, an every other-day ketogenic diet (EODKD) and a combination of KD with intermittent fasting has been introduced in order to combine KD with intermittent fasting as a new regimen for epilepsy therapy, with a better potential for seizure control [104]. In rats, EODKD determines higher ketone levels in both serum and cerebrospinal fluid than KD [105]. However, its effects on bone remain largely unknown. Recently, a study aimed to set different ketogenic rat models (thirty male Sprague-Dawley rats) and to compare the influence of EODKD with KD on bone microstructure and metabolism in KD treated rats, EODKD treated rats and controls, fed with continuous KD, intermittent KD and standard diet, respectively. After 12 weeks, BMD and body fat percentage were calculated by dual energy X-ray absorptiometry (DXA) scans, whereas the bone microstructure and mechanical properties were evaluated by micro-CT and a three-point bending test. Bone turnover parameters, specifically serum ALP and tartrate-resistant acid phosphatase (TRAP) were evaluated, together with the osteogenic capabilities of BMSCs according to both ALP activities and alizarin red stain results at different osteogenic stages. Either EODKD or KD treated rats showed higher ketone levels and fat percentage, with a lower body weight than the control group. Both EODKD and KD, however, resulted in compromising bone mass and mechanical properties. More specifically, EODKD demonstrated higher ketone levels and more pronounced inhibition of either the OCL process or early osteogenic differentiation than KD. Thus, if, on the one hand, EODKD accelerates ketosis, on the other hand it may not deteriorate bone microstructure or strength [106]. However, an interesting hypothesis, suggesting a bone compartmental-specific effect of KD, came from a study on 14 male Sprague-Dawley rats, which evaluated both the microstructures and mechanical properties of the skeleton. Rats, treated with KD or standard diet for 12 weeks, were equally divided into two groups. The KD regimen associated with higher ketone levels and lower glucose levels, with reduced body weight and total BMD. Bone morphometric analyses by micro-CT were performed at both cortical and trabecular bone of the middle L4 vertebral body, proximal humerus and tibia. Micro-finite element analysis (MFEA) provided the calculation of compressive stiffness and strength of the skeletal areas analyzed. After 12 weeks, the KD regimen associated with a reduction of BV fraction, Th.N of cancellous bone, cortical thickness, total cross-sectional area (tCSA) inside the periosteal envelope and of the cortical bone area of tibia and humerus. Contemporarily, Th.S was increased. Moreover, bone stiffness and strength were also found to be significantly decreased by the KD regimen, with a significant correlation with BMD and bone area at all the scanned sites. All this may suggest that KD leads to bone loss and a reduction in the biomechanical function more on appendicular bones than axial bones [107]. A subsequent study by the same research group, with similar methodological approaches, was performed on 48-weeks-old female C57BL/6J mice, firstly randomized into two main groups: (A) SD (standard diet); and (B) KD (ketogenic diet), then the SD and KD groups were subdivided into four groups, respectively: (1) SD + Sham (SD + ovary intact); and (2) SD + OVX (SD + ovariectomy); (3) KD + Sham (KD + ovary intact); and (4) KD + OVX (KD + ovariectomy). In addition to the previous study [108], in order to describe changes at the biomechanical and histomorphology level, a L4 and L5 vertebral compressive test and histological staining of these vertebrae were performed. An important reduction of the cancellous bone parameters was observed in KD + Sham and SD + OVX when compared to the SD + Sham group. Overall, the KD + OVX group exhibited the most serious bone loss. Qualitative parameter, such as the stiffness, significantly correlated trabecular parameters, resulted in the significantly higher SD + Sham group than all the others, with no difference observed between the remaining groups. The compressive test revealed a significant decrease in the failure load in the OVX + Sham and KD + OVX groups, but no difference was reported between the KD + Sham and SD + Sham groups. Such findings suggest that, similarly to OVX, KD may compromise the vertebral microstructure and compressive stiffness, thus indicating adverse effects of KD on the axial bone of these animals [108]. Another interesting study on mice models evaluated microarchitectural and biomechanical properties of KD-induced osteoporosis with and without the administration of the oral antidiabetic drug metformin (Met), comparing the effect of Met on the KD induced bone loss + OVX. As reported in the study, 40 female C57BL/6J mice were randomly divided into Sham, OVX, OVX + Met (100 mg/kg/day), KD (3:1 ratio of fat to carbohydrate and protein), and KD + Met (100 mg/kg/day) groups. After 12 weeks, bone mass and biomechanical parameters were taken in distal cancellous bone and femoral mid-shaft cortical bone. Biochemical parameters, such as the activities of serum ALP and TRAP, with the latter as a biochemical marker of OCL function and degree of bone resorption, were evaluated together with OC and TRAP immunohistochemistry staining. Both OVX and KD determined significant bone loss, compromising biomechanical properties in the cancellous bone, without an effect on cortical bone. The administration of Met enabled the increase of the cancellous BV/TV fraction following the OVX and KD, also enhancing the compressive stiffness following the OVX and KD. By a biochemical point of view, Met was associated with: (i) effectively increased serum ALP in the KD group; (ii) decreased serum TRAP in the OVX group; (iii) up-regulated expression of OC; and (iv) down-regulated expression of TRAP in both OVX and KD groups. The authors suggested that Met effectively attenuated the KD-induced cancellous bone loss, while it maintained the biomechanical properties of long bones. Table 4 summarizes the findings of all the above-mentioned studies. Overall, these data supported that KD enables the increase in the blood ketone level as well as the induction of the occurrence of osteoporosis through the impairment of the cancellous bone mass only in mice, similar to that determined by OVX. However, the use of Met as a potential treatment to prevent KD-induced osteoporosis in younger skeletons cannot be confirmed yet. [109]. Moreover, in rats treated with Low-carbohydrate high-fat diets (LC-HF), a lower rate of mesenchymal stem cells differentiating into OBLs has been reported. This finding may explain the observed LC-HF related reduced bone formation [110]. 

### 5.2. Clinical Studies on KDs and Human Bone Health

In the early 90s, a study showed analyzed stable serum ionized and total calcium, PTH and calcitonin levels in human subjects during a four-week long, very low caloric diet [111]. Subsequently, other data suggested that obese subjects undergoing a moderate VLCKD, but with high calcium intake (1200 mg/day), had a positive calcium balance. Importantly, the retention of ingested calcium was proportional to the CHO content of the diet [112,113]. However, scarce evidence exists on the fact that VLCD may play a negative effect on both BMD and BMC, in particular in the peripheral skeletal site with a higher percentage of cortical bone, proportionally to the rate of decrease of body weight, as well as in fat and lean mass [113,114].

#### 5.2.1. KDs, Growth Delay, and the Developing Skeleton

A reduced bone mass in children may be the result of inadequate bone accrual with decreased rat bone mineral deposition. Throughout all the childhoods, adequate bone mineralization verifies with a peak around the end of the adolescence time. Consequently, the amount of BMD achieved during childhood and adolescence will account for the risk of osteoporosis and fractures during adulthood [115]. Children not reaching adequate bone accretion have higher risk of fragility fractures (FFs), since, from childhood, they are also predisposed to develop involutional osteoporosis when adults [116]. Human studies on this issue are limited and are mainly performed on specific, restricted, peculiar populations such as the children affected by drug resistant epilepsy (DRE). Children with epilepsy, as well as other neurological disorders, are prone to low BMD and FFs [117,118,119]. It has been described that patients on KD had low BMD, with a significant association between low BMD and the duration of dietary therapy, and that KD might be associated with a growth delay. High ketone levels induced by high degree ketosis diets can be a potential cause of poor growth with a negative correlation between growth rate and ketone levels. After 12 months, 144 young subjects with high urinary ketone levels (80–120 mg/dl) were characterized by a reduced height versus those with moderate ketosis (<80 mg/dl) [120]. More recently, an Argentinian prospective study evaluated the growth and nutritional status in 45 children on KD, between 0.8 and 17.3 years of age. The nutritional follow-up was helpful to improve overweight and thinness, but showed growth deceleration in some of the subjects, confirming that children with refractory epilepsy treated with KD need careful growth monitoring. However, as was also declared by the authors themselves, several limiting problems affected the data interpretation of the study: (a) the lack of consideration of other causes of delayed growth, as represented by different antiepileptic drugs, type of epileptic syndrome, or severity of neurologic dysfunction; (b) the lack of addressing the association between height velocity and hormonal growth status before starting and while on KD [121].

Several studies explored the effect of the KD on the developing skeleton. Bergqvist et al. demonstrated a reduced bone mineral content (BMC) in patients on KD, with a follow up after 15 months. The study protocol was adequate according to its purposes, including questionnaires on calcium daily intake and DXA-assessed BMC performed at a sufficiently appropriate time interval for a correct interpretation of the DXA results themselves, even if BTMs were not included in the study [5]. A subsequent RCT with placebo on forty-two randomized subjects of both sexes analyzed the possible effects of two arms with different dietary treatments, and these lasted three weeks each, followed by a three-week washout period: (1) VLCKD1 arm, in which 50% of protein intake was replaced with synthetic amino acids; and (2) VLCKD2 arm, with placebo. The two VLCKD treatments (<800 kcal/day) differed in terms of protein content and quality. All the subjects were evaluated for various health parameters (health and nutritional status, by anthropometric analysis, DXA-assessed body composition, bio-impedancemetry, biochemical evaluation, and PPARγ expression by transcriptomic analysis), also including DXA-BMD and -BMC, before and after each dietary treatment. In this study, however, bone DXA scans, at baseline and at the end of each dietary treatment, were not appropriately performed with regard to the time interval (three weeks) between the two scans in order for the observed changes to be adequately interpreted. Moreover, the lumbar spine (LS) BMD was reported as T-scores (i.e., the BMD normalized for the BMD of a healthy young adults of the same population), but the population analyzed in both sexes was heterogeneous by age, ranging from 18 to 65 years, and consequently, the Z-scores (i.e., the BMD normalized for the BMD of individuals of the same sex and age of the same population) should have been reported in this population, especially because premenopausal women and male subjects younger than 50 years were included in the study [122]. Moreover, the daily calcium intake was not reported, thus preventing this study giving clear-cut conclusions. Another prospective longitudinal study was performed on 29 patients and was commenced on KD, for a minimum of six months, monitoring DXA-BMD and -BMC. Where possible, also areal BMD (aBMD) was converted to bone mineral apparent density (BMAD), and biochemical parameters, including serum calcium-phosphate, vitamin D, and BTMs, were collected. Patients were stratified for the level of mobility by the gross motor functional classification system and a trend towards a reduction in LS-BMD Z-score, 0.1562/year, was reported. Sixty-eight percent of patients exhibited lower BMD Z-score at the end of treatment. Interestingly, while less mobile patients showed lower baseline Z-scores, the bone loss rates were greater in the more mobile patients on KD. Importantly, height adjustment of DXA results was obtained for 13 patients, with 0.19 SD of mean reduction in BMAD Z scores. Approximately, 7% of these patients sustained fractures and the mean urinary calcium/creatinine ratio resulted in being generally elevated, even if only one patient developed nephrolithiasis. The authors found that children on KD showed differences in skeletal development, independent of height, potentially and possibly related to KD. In particular, such differences appear to be exaggerated in subjects who were ambulant [119]. A Swedish clinical study was performed on 38 pediatric subjects (55.2% girls) that were mainly affected by drug intractable epilepsy. They were treated by the modified Atkins diet (MAD) for 24 months. MAD is less restrictive than the KD, including 10–20 g of CHO per day, without limiting the amount of protein and as much fat as possible, and it is reported to be as efficacious as the classic KD in seizure reduction [123,124]. This study assessed growth, body composition, and bone mass in children, by DXA at baseline, at 6, 12, and 24-months visits, and serum calcium, phosphate, magnesium, alkaline phosphatase, cholesterol, 25-hydroxyvitamin D, IGF-1 and insulin like growth factor binding protein 3 (IGFBP3), at baseline and after 24 months. Approximately half of the patients treated with MAD showed a positive response to the diet, with over 50% seizure reduction. The authors concluded that MAD efficiently reduced seizures with no negative effect observed on longitudinal growth or bone mass, probably explainable also by the stable pH and lower 3βOHB level reached [125]. However, additional long-term studies concerning MAD and bone health are necessary. An eight-year retrospective observational cohort study, on 68 KD treated children with for more than six months, evaluated both the changes in DXA-BMD and the possible effect of an intravenous bisphosphonate therapy [anti-(bone)resorption therapy]. Fractures were reported in approximately 9.0%, and approximately 9.0% of the included subjects suffered from kidney stones. A statistically significant BMD increase, as compared to the baseline, has been reported in five patients treated with intravenous bisphosphonate therapy, while BMD did not improve in the non-pharmacologically treated group. This survey demonstrated that children on KD exhibited low to normal BMD values, susceptible to further decreases during KD, over the time they were under KD [126]. A speculative consideration can be made regarding the possible link between epilepsy associated with *GLUT1* gene mutations and the possible occurrence of unhealthy bone. It has been reported that vitamin C is transported as dehydroascorbic acid (DHA) in OBLs, as well as skeletal muscle cells, by functioning GLUT1 transporters. At physiological pH levels, DHA is uncharged and can easily diffuse through the plasma membrane, efficiently transported via GLUT1. Noteworthy, both insulin and IGF-1 stimulate vitamin C recycling, facilitating the accumulation of ascorbic acid (ASC) in OBLs [127]. Interestingly, insulin and IGF-1 increase the GLUT-mediated DHA absorption in OBLs by insulin-receptor-dependent mechanism, and then DHA is reduced to ASC, so that high ASC levels are available for adequate collagen synthesis [128]. All this amount of data suggests the hypothesis that in young *GLUT1* mutant epileptic subjects, an impaired vitamin C metabolism at OBLs level may influence both an abnormal synthesis and production of collagen with a consequent negative impact on bone mineralization and strength that could be clinically manifested at a later age in life.

#### 5.2.2. KDs and Calcium Metabolism

A high incidence of renal calculi with hypercalciuria have been reported in patients on KD, suggesting a major effect of the KD on calcium metabolism [129,130]. KD-related calcium and bone mass deficiency were originally reported in five children maintained chronically on combined KD-anticonvulsant drug therapy [131]. While studies on these children suggested that chronic adaptation to a ketogenic Low CHO High Fat (KLCHF) diet may be associated with poor bone health [5,110,132,133], it still remains unknown if changes in BTMs persist or amplify after chronic exposure to low CHO availability. Anti-epileptic drug (AED) use associates with lower BMD in both adults and children [134,135,136,137], and epileptic adults, who have been treated with AED since childhood, exhibit lower bone mass than epileptic adults who started AED therapies in adult life. These findings point out a specific effect of AEDs on the developing skeleton [138], as also to a possible cumulative effect. Over the past 15 years, KDs represent a valid therapeutic intervention for these patients [139,140]. A systemic ketosis can be obtained through the maintenance of a high fat, low CHO, and restricted protein intake, with a beneficial effect on raising seizure threshold [141]. However, a chronic ketoacidosis state may create the conditions for both a negative balance of bone mineral ions (buffering capacity) and a decreased renal conversion of 25OHD to its active metabolite 1,25(OH)_2_D [142]. Of course, the presence of cerebral palsy as comorbidity further increases the risk of BMD decrease likely through an increased bone turnover during a ketonic state in epileptic adults [143].

### 5.3. Duration of the KD Regimen and Study of Bone Health Parameters

The use of KD therapies in adults has expanded in the last two decades. In general, long-term KDs are “intended” for adult patients with epilepsy, while shorter-term KDs (usually lasting 3–4 months, less frequently than six months or more) are used in patients suffering from various dysmetabolic conditions (e.g., T2DM, obesity, PCOS and so on), even if they can be repeatedly performed over time. While possible correlations between KD and bone health were at least conceivable in children, it remains unclear whether the KD effects on bone health may be different in adults. In any case, available data are derived from studies mainly on adults undergoing KD, similarly to what was reported for children, with most studies being not adequately designed to provide relevant data on bone involvement [144,145,146,147]. More recently, data on other peculiar human subjects under “temporary” restricted CHO regimen or KD, such as professionals and non-professional athletes, are emerging. An acute increase of bone resorption in CHO restriction, before, during, and after prolonged endurance exercise has been reported [148,149,150]. This increased bone resorption has been hypothesized to be potentially linked to the concomitant increases in IL-6 concentrations [148]. However, acute reductions in CHO availability have been suggested to modulate the increase of bone resorption markers independently of the energy availability and circulating IL-6 levels [151]. Increased post-exercise IL-6 concentrations have been described in elite race walkers following a 3.5-week adaptation to a ketogenic low-carbohydrate, high-fat (LCHF) diet [152]. A study on short term LCHF diet regimen aimed to investigate the interactions between diet/exercise and BTMs [cross-linked C-terminal telopeptide of type I collagen (CTX), procollagen 1 N-terminal propeptide (P1NP), and OC] in thirty world-class race walkers (25 males and 5 females) during a 3.5-week adaptation to a LCHF diet, followed by restoration of CHO feeding. The BTMs were evaluated at rest at baseline, after the 3.5-week intervention, after acute CHO feeding (fasting and 2 h post meal) and after exercise (0 and 3 h). The authors found BTMs decrease after short-term LCHF diet, with only one resorption marker recovering after acute CHO restoration [151]. However, long-term studies of the effects of LCHF on bone health are necessary. Table 5 summarizes most of the cited human clinical studies.

### 5.4. VLCKD and Acid-Ash Proteins Diet

While no studies have been carried out in VLCKD, diet high in acid-ash proteins (acid-ash hypothesis identifies protein and grain foods as cause of the calcium release from the skeleton to buffer the acid load from the diet and increase urinary excretion of calcium) [153] has been associated with excessive calcium loss due to its acidogenic content. Calcium has provided as a buffer from the skeleton through the active resorption of bone; indeed, calciuria is directly related to net acid excretion and it is not compensated by increasing intestinal calcium absorption [154]. Thus, taken together, all these observations raise some concerns about the risk of loss of BMC during VLCD. To prevent such a consequence, it is recommended to provide an adequate high intake of calcium and vitamin D, as well as an appropriate amount of CHO.

### 5.5. KDs and the Role of Mineral Supplements

Apart the protective role of KDs/KBs on the epileptic clinical manifestations, it is also important to stress the importance that these diets are adequately associated with appropriate supplements of minerals, such as calcium, preferably as calcium citrates, and magnesium salts [138,155], and vitamins, including C and D, in order to confer a sort of skeletal protection, especially, but not limited to, young subjects destined to follow KD regimens for long periods of time, if not for all the life. In particular, associated treatment with potassium citrate may act as a protecting factor to prevent nephrolithiasis due to aciduria and hypocitraturia [155] induced by KD and MAD. Of course, randomized and controlled clinical trials versus placebo are not ethically recommendable in these patients, but both in vitro and in vivo model studies on animal/human mature bone/osteoprogenitor cell/primary culture cells may give further insights on this topic. Finally, an important question is still open: how much might KBs support these different aspects of bone pathophysiology?

## 6. Some Specific Key Points to Be Considered to Adequately Perform Clinical Study Having Bone Health as Primary Endpoint in Subjects Treated by KDs, VLCKD

Clinical studies concerning the potential effects of KDs on human bone health are scarce and suffer from experimental designs and methodological approaches, which render these studies not always adequate for the intended purposes. Moreover, not many data are available on long-term fracture risk in individuals undergone KDs.

In general, especially for studies investigating the effects of KDs and VLCKD in children, the sample size of all these studies is limited. In children studies, moreover, BMC is the preferred outcome measure since, in this age range group, the bone expands and the BMC increase at different rates during childhood [156], and in order to combine DXA results for individuals of different ages and to adequately “correct” for the growth-related changes in BMC, Z-scores for BMC/age and BMC/height need to be calculated based on a healthy reference sample. Height adjustment of DXA results, such as aBMD converted to BMAD, should be performed when it is possible [137].

### 6.1. The Importance of Least Significant Change Concept

The concept of least significant change (LSC) is calculated by the formula 2.8× precision error. Indeed, the LSC commonly recommended for detecting a variation in a subject with only a 5% chance of a type 1 error (i.e., stating that there is a variation when in reality there is not) is 2.8 times the inter-assay coefficient of variation of the diagnostic test used. In other words, the LSC represents the smallest difference between two f BMD or BTMs measurements that could be considered as a real variation and not attributable to chance. The LSC is fundamental when a clinical study on bone health is designed. For example, BMD changes are considered significant if they are above the least significant change, that for LS and total hip BMD are generally about 2.8% and 4.8%, respectively and for BTMs may reach the 30%Consequently, the minimum time interval to elapse between the assessments of BMD or BTMs must also be calculated by taking these aspects into consideration [157].

#### 6.1.1. DXA Measurements

Except for special cases (e.g., first check after diagnosis and start of gluten-free diet in celiac subjects, patients with severe hypercortisolism or hyperparathyroidism), it is not appropriate to repeat the DXA scan before 15–18 months have passed. Indeed, since the BMD/BMC changes detected in the short term could be less than the margin of error of the measure (coefficient of variation) (www.iofbonehealth.org). Moreover, when the population study is a heterogeneous population by age, including both sexes, ranging from ≤18 to ≥65 years, Z-scores should have always to be evaluated in premenopausal women and male subjects ≤50 years of age. In case of KD-treated obese individuals, a study on these patients revealed that >50% of the included subjects, with at least one vertebral fracture, had a normal or only a slightly reduced BMD, but not osteoporotic values, with 4.4-fold higher vertebral fractures occurrence than in controls. Then, in obese subjects, DXA may not represent an accurate instrument to adequately estimate the fracture risk [158,159], especially when DXA-based BMD assessment is performed only at the lumbar and femoral sites. In this case, the BMD evaluation at the level of non-dominant forearm can be more useful than the previously mentioned skeletal sites [160].

#### 6.1.2. BTMs Measurements

Similar recommendations have to be done also regarding the clinical use of BTMs, such as ALP, carboxy-terminal collagen crosslinks (CTX), and procollagen type I N propeptide (PINP). All these BTMs have important analytical and preanalytical variability, with differences depending on the specific BTM considered, the assay used, and the technician’s expertise [157]. A clinical study on postmenopausal osteoporotic women, randomized to alendronate, ibandronate or risedronate (all amino bisphosphonates), plus calcium and vitamin D supplementation for 2 years, revealed the existence of an LSC of 56% for CTX and 38% for PINP, respectively [161]. Recently, the International Osteoporosis Foundation (IOF) and the European Calcified Tissue Society (ECTS) working group published recommendations specifically limited to the use of BTMs in the screening of the adherence to oral therapy with bisphosphonates, indicating a period of at least three months between basal, before starting oral bisphosphonate therapy, and follow-up monitoring of CTX and/or PINP [162]. Therefore, currently, all these recommendations for an appropriate time interval to elapse between DXA and BTMs measurements must also be followed in clinical protocols searching for correlations between KD/VLCKD regimens and bone health. Finally, in pediatric populations undergoing KDs, MADs, the lack of age- and sex-, diagnosis-, height-, BMI-, bone age- and puberty-matched the control group represents a strong limitation.

### 6.2. Other Suggestions

Specific food intake questionnaire, for example concerning the calcium/phosphate daily intake, the daily sun exposure should be always included at the basal and during follow-up visits in these studies, together with phospho-calcium metabolism investigations (at least 25-hydroxyvitamin D (25OHD), serum calcium, serum phosphate, BTMs, citraturia, calciuria and phosphaturia in 24 h urine collection), and sex hormonal status (post pubertal age). A stratification for the level of mobility by appropriate functional classification system should be included in the study design and all the environmental and pathological factors, concomitant therapies, in addition to other factors negatively impacting/influencing the bone health, which should be carefully considered and excluded and/or statistically evaluated to give weight-adjusted results. Table 6 provides, as a general example, suggestions on clinical, biochemical, and instrumental tests to predict bone health-related endpoints in various KDs regimen lasting 24-months protocol. Clinical trials of less than 12 months duration are not really appropriate to evaluate skeletal endpoints by a statistical point of view. All the other types of tests to be potentially studied for this purpose are not considered here, but each researcher will establish them on the basis of the other non-skeletal endpoints and the type of population under study.

## 7. Inflammation, Diet, and Bone Health

Inflammation may be closely related to bone metabolism, in particular to the remodeling phase, and consequently, to the pathophysiology of postmenopausal osteoporosis. In fact, interleukin (IL) molecules, IL-1, IL-6, and tumor necrosis factor-alpha (TNF-α), all known pro-inflammatory cytokines, may regulate bone resorption [164,165]. Estrogens attenuate oxidative stress and the differentiation and the apoptosis of OBLs [166] and the postmenopausal estrogen deficiency correlates with increased monocytic production of IL-1. Increased circulating TNF-α levels are found in women affected by severe (or that presence of fragility fracture) osteoporosis [167]. Inflammatory process also produces free radicals, metabolites of arachidonic acid that, together with other cytokines, and chemokines, contribute to the further production of free radicals, such as ROS [168], which are known to be involved in the apoptosis of OBL and in OCLgenesis, and therefore, bone resorption [166]. In mice, the oxidative stress demonstrated to antagonize the Wnt pathway, necessary for a corrected OBLgenesis from precursors [166], and the age-related oxidative stresses may be fundamental in the decline of human bone mass and strength [169]. Interestingly, dietary factors may modify the inflammation, and associations between dietary inflammatory index (DII)^®^ and inflammatory markers have been investigated [170]. CHO, saturated fatty acids, proteins, and other micronutrients represent pro-inflammatory dietary factors that should be adequately counterbalanced by contents of anti-inflammatory micronutrients, such as low unsaturated fatty acids, flavonoids, vitamin A, vitamin C, vitamin D, vitamin E, and selenium. Some studies suggested a significant association between DII^®^ and bone health as reported in an Iranian women population where higher DII^®^ scores more likely had below-median BMD values at lumbar spine [171]. In the Women’s Health Initiative data longitudinal study, women with the least inflammatory dietary patterns exhibited greater lower-hip BMD loss [172]. While an increase in osteoporosis risk has been described in postmenopausal women with an increase in DII^®^ [173], a contemporary cross-sectional analysis of DII^®^ and BMD in young adult subjects failed to find such correlation [174]. Therefore, further clinical, observational, and basic research studies, possibly not limited only to postmenopausal women, have to be developed to better define the relationship between diet micronutrients mixture components, DII^®^ and bone health, and to describe what kind of relationship it is.

## 8. Future Perspectives on KDs, Microbiota, microRNAs, Bone Marrow Lipids Composition, and Bone Health

Studies on the possible interactions between KD regimens, gut microbiota, microRNAs (miRNAs), and bone health may offer new opportunities to better understand the therapeutic potentials of these dietary therapies on distinct cells, organs, or at the systemic level. In the last ten years, a large body of evidence revealed the importance of the human gut microbiota on human health, being involved in several and different pathophysiological functions [174,175]. Unfortunately, the high variability of microbiome composition among people, makes it currently difficult to identify the various microbiota changes in relation to a specific dietary pattern including KDs [176].

### 8.1. Microbiota, Short-Chain Fatty Acids, Butyrate, and Bone Health

The gut microbiome has been proposed as a key regulator of bone health, potentially influencing both the postnatal skeletal development and the skeletal involution. Abnormal changing in microbiota composition and host responses to the microbiota has been recently suggested to sustain pathological bone loss. Prebiotics and probiotics nutritional supplements can determine changes in microbiota composition enabling the prevention, or the reversion, of bone loss by the production of metabolites, such as short-chain fatty acids (SCFAs), by the gut microbiota [175]. Animal studies strongly suggested that SCFAs might blunt osteoclastogenesis and bone resorption, stimulating bone formation. In particular, the probiotic Lactobacillus rhamnosus GG (LGG) was demonstrated to increase bone mass in mice through the butyrate production, precursor of β-hydroxybutyrate, and LGG or butyrate contribute to expand the pool of Treg cells, both in the gut and the bone marrow. These cells enable the upregulation of the osteogenic Wnt ligand Wnt10b expression by CD8^+^ T-cells and Wnt10b may stimulate bone formation by activating Wnt signaling in OBLs, supporting the hypothesis that the bone anabolic activity of SCFAs depends on Tregs and CD8^+^ T-cells [176], even if the antiresorptive activity of SCFAs could be T cell–independent [177,178]. The lowering of butyrate levels by microbiota depletion alters the requirement for PTH in stimulating bone formation and bone mass increase, and the reestablishment of physiologic levels of butyrate also restored the PTH-induced anabolism. Thus, the gut luminal microbiota-derived butyrate may play an important role to positively stimulate the regulatory pathways underlying the anabolic action of PTH in bone [179]. In mice models of primary and secondary hyperparathyroidism, the presence of segmented filamentous bacteria (SFB) in the gut microbiota enables the PTH-dependent expansion of intestinal TNF positive (TNF^+^) T- and Th17-cells, increasing their egression, by sphingosine-1-phosphate (S1P)-receptor-1 mediated mechanism, from the gut and their recruitment to the bone marrow, determining bone loss [180]. Finally, a plasma high-resolution metabolomics approached cross-sectional study on 179 adult animals, suggest that either pro-inflammatory fatty acids or gut microbiome may be regarded as novel regulators of postnatal bone remodeling [181]. All these interesting studies need also need the application on humans.

### 8.2. KDs, miRNAs, and Bone Health

Circulating microRNAs (miRNAs) may represent “new” potential biomarkers for the early diagnosis of bone metabolism impairment [182]. Their diagnostic potential emerged in several human oncological, dysmetabolic, and cardiovascular pathologies [183,184,185,186,187,188,189,190], and also in bone metabolism diseases, such as primary, secondary, and idiopathic osteoporosis, suggesting a possible correlation between the molecular signature of circulating miRNAs and osteoporosis. Recent studies also investigated their potential role as biomarkers for osteoporosis diagnosis and osteoporotic fracture prediction [191,192,193,194,195,196,197,198]. A case-control study on 448 individuals, subdivided in normal, osteopenic, and osteoporotic cohorts, suggested that a panel of specific circulating miRNAs may represent a potential biomarker for osteoporosis, also enhancing the osteoporosis detection rate as a method complementary to DXA [199]. KD has been reported to regulate the microRNA expression profile in obese subjects [200,201,202,203,204], and study identified a panel of 11 miRNAs normalized by KD regimen when compared to the lean counterpart, being also able to counteract inflammatory and oxidative stress, regulating the ROS-producing molecular network [205]. In a study of 36 obese subjects, equally distributed in both sexes, undergoing six weeks of biphasic KD, some miRNAs have been identified as potentially useful to monitor low CHO nutritional regimens. At the same time, specific miRNAs panels, with a cell- and/or tissue-specificity and/or in a health status-dependent manner, could be identified as opening the possibility to use them as biomolecular/biochemical tools, indirectly reflecting the regulatory biochemical mechanisms underlying cell signaling pathways. These preliminary results are encouraging and may also represent a fertile ground for new pathophysiological knowledge of bone metabolism for both diagnostic and therapeutic purposes, although neither basic nor clinical studies have not yet been published, so far.

### 8.3. Bone Marrow Lipids Composition and Bone Health

The bone marrow lipid composition does not strictly depend only on lipids dietary intakes and may act in bone homeostasis as a key regulator. Palmitic and stearic saturated fatty acids may result in being lipotoxic to bone [206,207] and impair the OBL physiology [208,209], whereas palmitoleic, oleic, and linoleic unsaturated fatty acids seem to promote OBL differentiation, survival, and mineralization activity [206,209,210], while inhibiting OCL both differentiation and function [211,212]. Considering the association with bone fragility [213,214], it has been suggested that bone marrow saturated index, the proportion of saturated fatty acids to total fatty acids, to be a potential biomarker for osteoporosis [215]. By a lipidomic approach in a pediatric population affected by varying degrees of bone fragility, the investigation of bone marrow lipidic composition was revealed to be a test that may amplify the clinical armamentarium of measures powering both research and clinical studies [216,217,218,219]. Thus, an appropriate dietary intake of lipids has to be maintained for not only cardiovascular health, but probably also for skeletal integrity even if further appropriate in vitro and clinical studies in humans are needed.

## 9. Conclusions

In summary, at present there are no human clinical studies with adequate powerful experimental design to definitively determine if subjects under KD therapies can derive positive/negative effects on bone health. Apart from the subjects with AED epilepsy, either *GLUT1* mutant or not, most of the patients undergone KDs are exposed to these regimens for periods of time probably not long enough to create qualitative/quantitative bone damages or to be currently clinically and/or biochemically perceived. Clinicians should also be aware of the potential skeletal side effects during KD treatment, and consequently, monitor bone health status. Longer term follow-up is required to determine peak bone mass, bone mass maintenance and fracture risk throughout life, consequently adequate and appropriate monitoring of BMD and BTMs has to be considered crucial, as well as the monitoring of occurrence of kidney stones and hypercalciuria. Multidisciplinary taskforces, including the bone specialist, are needed in order to adequately design and power clinical studies that are able to evaluate the potential effects on the bone health of various KD schemes.

## Figures and Tables

**Figure 1 ijms-22-00435-f001:**
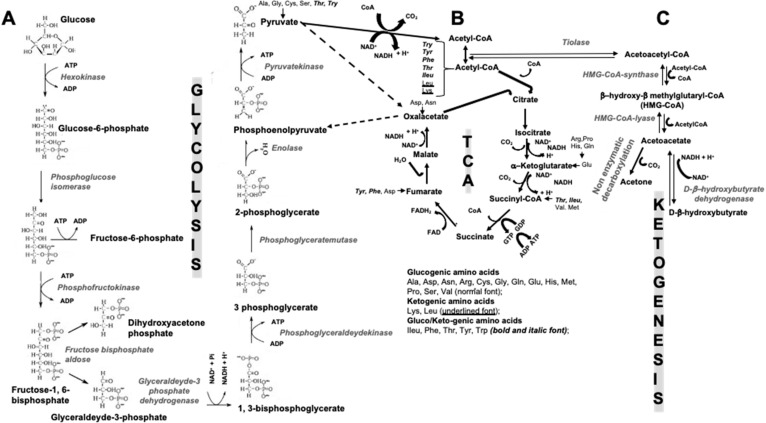
Scheme of interconnections between glycolysis (**A**), tricarboxylic acids cycle (TCA) (**B**), and ketogenesis (**C**). ADP = adenosine diphosphate; ATP = adenosine triphosphate; Acetyl-CoA = acetyl coenzyme A; CO_2_ = carbon dioxide; CoA = coenzyme A; FAD = flavin adenine dinucleotide; GDP = Guanosine diphosphate; GTP = Guanosine triphosphate; H^+^ = positively charged hydrogen ion, i.e., a hydrogen atom deprived of its electron; HMG-CoA = 3-hydroxy-3-methyl-glutaryl-coenzyme A; NAD = nicotinamide adenine dinucleotide; NADH = reduced version of NAD; Pi = inorganic phosphate; at the bottom of Figure 1B (under the TCA) classification of glucogenic-, ketogenic-, and gluco/ketogenic amino acids is reported together with the specification of the font type in which they are written.

**Figure 2 ijms-22-00435-f002:**
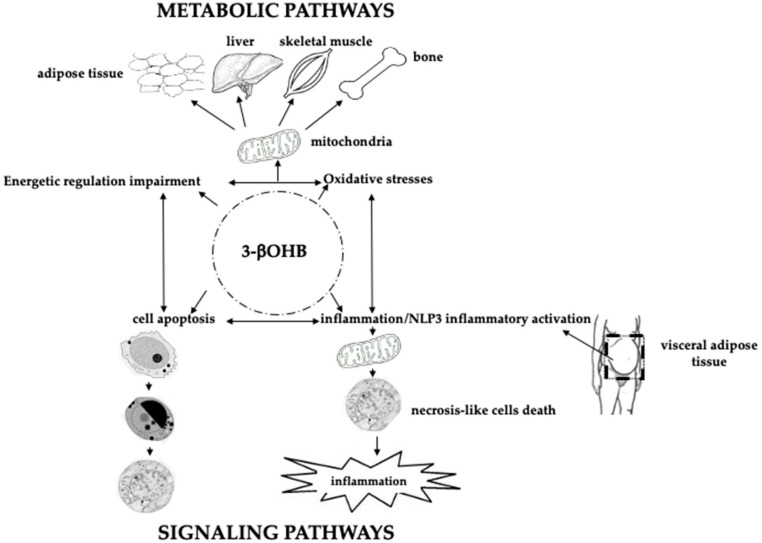
Schematic role of 3betahydroxybutyrate (3βOHB) in energy metabolism, response to oxidative stress, apoptosis, inflammation and signaling pathways. In the center of the figure, within the dotted circle, is depicted the molecule of 3βOHB, “central” to all the physiological events described. The top of the figure shows how 3βOHB, “through the intervention of mitochondria”, may impact the metabolism of adipose tissue, liver, skeletal muscle, and bone, both in terms of oxidative stress modulation and energetic regulation impairment. In turn, oxidative stress, and energetic regulation impairment impact, even with 3βOHB, on cell apoptosis, and inflammation/NLP3 inflammatory activation, (lower figure). Visceral adipose tissue plays an active role on the above events. Thus, 3βOHB may act as a signaling molecule.

**Figure 3 ijms-22-00435-f003:**
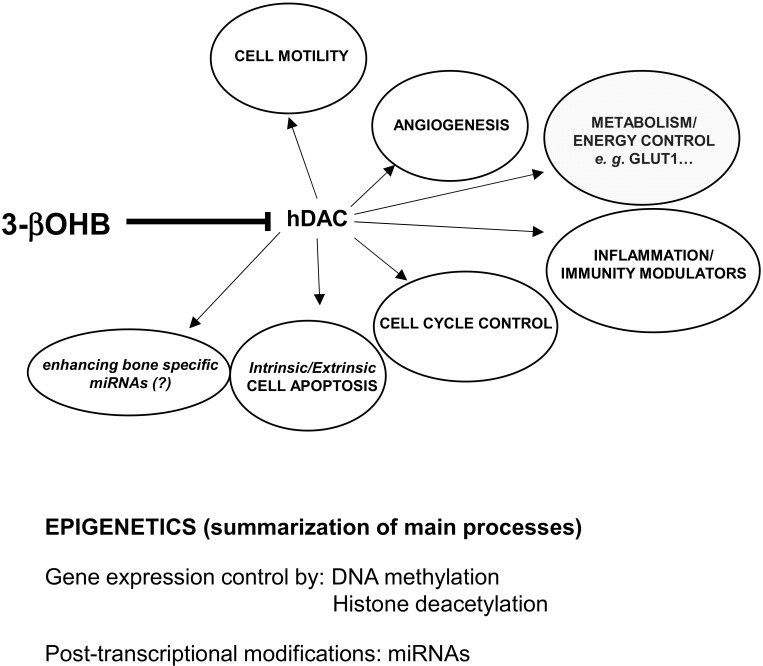
Molecular links between biological functions, environmental factors and 3betahydroxybutyrate (3βOHB). 3βOHD inhibits the function of histone deacetylases (hDAC), molecules representing key transcriptional cofactors that regulating gene expression by deacetylation of lysine residues on histone and nonhistone proteins. Activity of hDAC is important to regulate/influence all the physiopathological events inserted inside the ellipsoidal figures. Through the influence exerted on the processes of DNA methylation, histonic deacetylation and microRNAs expression, 3βOHB may act at epigenetic levels, mostly involving changes that affect gene activity and expression.

**Table 1 ijms-22-00435-t001:** Endocrine effects/functions of UcOC, mediated by binding to specific GPRC6A membrane receptor [15], on insulin, energy, and testosterone metabolism.

Effects on Insulin Metabolism	Effects on Energy Metabolism and Other Suggested Endocrine Effects
Increase of β-cells proliferation, synthesis, and secretion of insulin	Improvement of energy expenditure through multiple mechanisms [15].
Improvement of insulin sensitivity through multiple mechanisms [15].	Increased expression of adiponectin (ADPN) in white fat and reducing lipid accumulation (and inflammation in the steatosis liver).
Modulation of hepatic insulin sensitivity	Stimulation of energy expenditure by increasing mitochondrial biogenesis in muscle and regulating the expression of genes involved in energy consumption in brown adipose and musculoskeletal tissue.
	Promotion of male fertility by stimulating the synthesis of testosterone in Leydig cells.

**Table 2 ijms-22-00435-t002:** Scheme of the main biochemical processes during osteoblast differentiation, related to “environmental” O_2_ conditions.

Low O2 Conditions	Normal O2 Conditions
Glucose is converted into pyruvate with ATP production, and pyruvate is converted into lactate. Then, 80% of glucose is converted into lactate, representing the “predominant destiny” of glucose in OBL	Pyruvate enters the mitochondria where it is oxidized to CO2 + H2O

**Table 3 ijms-22-00435-t003:** IGF-1 levels and their correlation to 3βOHB blood levels in famine, and IGF-1 levels KDs relationship.

IGF-1, 3βOHB, and Famine	IGF-1 and KDs
IGF-1 levels reduce into famine [87].	IGF-1 levels may be reduced up to 46% after 7 days of KD [88].
3βOHB blood levels and IGF-1 related growth speed inversely correlated [89].	KD-induced famine-like metabolic state may reduce either the IGF-1 levels or its bioavailability [89,90].

**Table 4 ijms-22-00435-t004:** Summary of the findings of animal studies related to KD and bone health.

Model	Evaluated Parameters	Findings/Conclusions
Forty C57BL/6J 8-week-old female mice randomly divided into SD + Sham, SD+OVX, KD + Sham KD + OVX groups; KD fed for 12 weeks [103]	Micro-CT scanning of distal femur trabecular bone and middle femur of cortical bone; tibial maximum bending force and stiffness; TRAP, collagen type I (ColI) expression, and OC staining	*KD compromised cancellous and cortical bone of long bones. Combination of KD and OVX may lead to more bone loss.*
Sprague-Dawley rats control, KD and EODKD groups, fed with standard diet, continuous and intermittent KD, respectively [106]	ketone levels	*EODKD demonstrated higher ketone levels than KD, inhibition of either osteoclastic process or early osteogenic differentiation.*
	DXA-assessed BMD and body fat percentage; micro-CT analysis of bone microstructure and mechanical properties, three-point bending test; bone turnover markers (serum ALP and TRAP), plus osteogenic capabilities of BMSCs (ALP activities and alizarin red stain) in different osteogenic stage.	
Sprague-Dawley male rats KD or a standard diet for 12 weeks, equally divided into two groups [107]	Cortical and trabecular bone micro-CT morphometric analyses; micro-finite element analysis to calculate compressive stiffness and strength of the skeletal areas.	*KD leaded to bone loss and reduction of the biomechanical function more on appendicular bones than axial bones*
48-weeks-old female C57BL/6J mice Two main groups: A) SD; and B) KD; SD and KD were then subdivided into four groups: A1) SD + Sham (SD + ovary intact); and A2) SD + OVX (SD + ovariectomy); B1) KD + Sham (KD + ovary intact); and B2) KD + OVX (KD + ovariectomy) [108]	L4 and L5 vertebral compressive test and histological staining of these vertebrae	*The compressive test decreased in the failure load in OVX + Sham and KD + OVX.*
female C57BL/6J mice Randomly divided: Sham, OVX, OVX + Met (100 mg/kg/day), KD (3:1 ratio of fat to carbohydrate and protein), and KD + Met (100 mg/kg/day) groups [109]	After 12 weeks: bone mass and biomechanical parameters in distal cancellous bone and femoral mid-shaft cortical bone; biochemical parameters: serum ALP and TRAP, OC, and TRAP immunohistochemistry staining	*Met associated with increased serum ALP in the KD group, decreased serum TRAP in OVX group, and up-regulation and down-regulation of OC and TRAP expression, respectively.*
Twenty-four male Wistar rats fed for 4 weeks either normal chow (CH, 9% fat, 33% protein, and 58% carbohydrates), LC-HF-1 (66% fat, 33% protein, and 1% carbohydrates), or LC-HF-2 (94.5% fat, 4.2% protein, and 1.3% carbohydrates) [110]	pQCT, µCT analysis; three-point bending test; histology of humeri, bone marrow quantitative real-time PCR; P1NP, OC, PINP1, CTX, IGF-1, 25OHD, GH, leptin, IGF-1, Insulin-like growth factor binding protein 2-3, rat intact PTH, serum calcium and phosphate	LC-HF diets associated with more visceral and bone marrow fat, increased leptin but decreased IGF-1 and body length; LC-HF-2 reduced humerus, tibia, and femur lengths; pQCT and µCT revealed significant reductions in tibial BMD in both groups; in LC-HF groups, tibial maximum load was impaired and serum PINP1 reduced; in LC-HF, real-time PCR showed bone marrow reduced expression (70 to 80%) of Runx2, osterix, and C/EBPβ.

**Table 5 ijms-22-00435-t005:** Summary of the findings of human clinical studies related to KD and bone health (systematic reviews and metanalysis are not included).

Population	Type of Study	Outcome Considered and Methods to Evaluate Diet and/or Bone Health Parameters	Findings/*Conclusions*
Twenty-five US children with intractable epilepsy (IE), on a 4:1 KD (by weight fat:CHO and protein) [5]	2-phase, 15-mo longitudinal study of growth and nutritional effects of KD	Effect of KD on bone health, growth, nutritional status, and seizures; DXA-assessed whole-body and LS-BMC z score; evaluation/measurements of demographics, anthropometry, serum 25OHD, intact PTH, electrolytes, and dietary intake. All at baseline and at 3, 6, 12, and 15 mo.; evaluation of BMC change over time.	*Bone health was poor, particularly for younger nonambulatory children with low BMI status. KD resulted in progressive loss of BMC*
Eight UK obese subjects on therapy with low energy (2741–3301 kJ/day), high calcium (28.9–35.1 mmol/day) diets [112]	Observational	Effects of dietary CHO content and of tri-iodothyronine administration on calcium, zinc, and phosphate balances and on urinary hydroxyproline output	*Calcium balances generally positive at the high levels of intake, but significantly more calcium was retained when dietary CHO intake increased*
Twenty-one US healthy obese women aged 38 +/− 9 years, randomly on either diet alone or diet plus resistance training. Both groups with a 925-kcal/d portion-controlled diet for the first 16 of 17 weeks and a 1000 to 1500-kcal/d balanced deficit diet thereafter [113]	RCT	DXA-BMC, -BMD, fat-free mass (FFM), and DXA-fat mass before and after 24 weeks of dieting	*Diet and resistance training not associated with a better outcome vs. diet alone. Increasing the energy content of very-low-calorie diets to 925 kcal/d may prevent the loss of total BMD, but not the loss from the FN and GT*
Two hundred and seven healthy Caucasian, aged 9–18 yrs. [115]	Cross-sectional study	Rate of skeletal growth at LS-BMD-BMC and FN-BMD-BMC sites, in relation to age and pubertal stages in both sexes	*Marked age-related delay in L2-L4-BMD or -BMC increase in males vs. females, unrelated to pubertal stages; higher mean values in males for BMD and BMC, at the end of the rapid growth spurt; dramatic reduction (2–4 years after menarche), in bone mass growth in females >15 yrs. of age*
One hundred and seventeen US subjects, from 3-county areas, age range 2–19 years with moderate to severe cerebral palsy (CP) according to the Gross Motor Functional Classification scale. Population-based sampling from both hospital- and school-based sources [117]	Observational study	DXA assessed LS- and distal femur-BMD; anthropometric assessment of growth and evaluation of nutritional status; Child Health Status Questionnaire; serum calcium, phosphate, ALP (total and bone-specific), OC, NTx, total protein, albumin, prealbumin, 25OHD	*Femur osteopenia in 77% of the cohort and 97% of all study participants.* *Fractures in 26% of the children older than 10 years.* *Severity of neurologic impairment, increasing difficulty feeding, use of anticonvulsants, and lower triceps skinfold z scores, independently contribute to lower femoral BMD z scores. Low BMD prevailed in moderate-severe CP and associated with significant fracture risk*
Fifty-seven US subjects, age range 1-26 years, on KD for seizures management and one on KD for GLUT1 deficiency. Fourteen children had CP other than seizure disorder [120]	Retrospective study analysis	3-day food diary nutrient analysis. Height or length and weight at 0, 6 mo., and 12 mo. follow-up visits.	KD for 12 months significantly decreased height z scores. Daily urinary ketone levels (reagent strip) categorized into two groups: <80 mg/dL (moderate ketosis) and 80–120 mg/dL (high ketosis). High ketosis significantly decreased height z scores. *Subjects on the KD showed a delay in growth*
Fourthy-five Argentinian children, 23 boys, on KD for at least two years, between 0.8 and 17.3 years of age (mean 6.6) [121]	Prospective cohort	Growth assessment (weight, height, and BMI). Standard deviation scores for all measurements calculated at KD initiation and at follow-up	*Height growth < percentile third for 2 boys and 2 girls*
Sixty-three Australian patients on the KD, 29 patients on the KD for a minimum of 6 months (range 0.5–6.5 years, mean 2.1 years) [109]	Prospective, longitudinal study	LS-, right TH-DXA z-scores at baseline and 6 mo. intervals; serum calcium, phosphate, 25OHD, PTH, ALP, OC, urine calcium/creatinine ratio at baseline and 3 mo. intervals; gross motor functional classification system (GMFCS)	*Trend in reduction of LS-BMD z-score/year; 68% had lower BMD z-score at the end of treatment. Lower baseline z-scores in less mobile patients, but the rate of bone loss on the diet was greater in the more mobile patients. Only 2 patients fractured.* *Elevated mean urinary calcium-creatinine ratios; 1 patient developed renal calculi*
Seventy-five UK children providing growth data on 1 of 2 KDs at baseline and after 3, 6, and 12 months, if continued [141]	Prospective, longitudinal study	Weight, height, and BMI z scores at baseline and 3, 6, 12 mo.; evaluation of growth trend.	*Height z scores decreased at 6 and 12 mo., particularly in younger and ambulatory children; weight z scores decrease in the MCT group only at 3 and 6 mo. and in both groups at 12 mo.; forty children completed the study with no differences in growth trend between classical and MCT diets for weight, height, or BMI; the MCT group had significantly higher protein intake*
One hundred ninety-five US children on the KD for intractable epilepsy from 2000 to 2005 [142]	Cohort study kidney stones formers vs. not kidney stones formers	Demographics, urine laboratory markers, and intervention with urine alkalization (potassium citrate)	Thirteen children developed kidney stones; oral potassium citrate significantly decreased the prevalence of stones and increased the mean time on the KD before a stone was first noted; the prevalence of kidney stones trended toward higher correlation with hypercalciuria (92% vs. 71%). *No child stopped the diet due to stones; kidney stones continue to occur in approximately 1 in 20 children on the KD*
Thirty US patients (15 study subjects and 15 controls). The 15 patients on diet to consume less than 20 g of CHO per day for the 1st month and then less than 40 g per day for months 2 and 3. Control subjects without restrictions on diet [45]	3-month study	Primary end point: urinary UNTx at 3 mo. Secondary end points: UNTx at 1 mo., BSAP at 1 mo., bone turnover ratio (BSAP/UNTx) at 1 mo., and weight loss	*Patients on low-CHO diet lost significantly more weight than controls, the diet not increased bone turnover markers vs. controls.* *No significant change in the bone turnover ratio compared with controls*
Thirty-eight Swedish patients, mean age (SD) 6.1 years (4.8 years), 21 girls, with intractable epilepsy, glucose transporter type 1 deficiency syndrome, or pyruvate dehydrogenase complex deficiency [125]	Prospective longitudinal cohort study	Assessment of growth, body composition, and bone mass in children on MAD for 24 months. Body weight, height, BMI, DXA-BMD, serum calcium, phosphorus, magnesium, ALP, cholesterol, 25OHD, IGF-1 and IGFBP3 at baseline and 24 mo. of MAD	*Approximately, 50% of the patients responded with more than 50% seizure reduction. Weight and height standard deviation score stabilized over 24 months; median BMI SDS significantly increased.* *MAD efficiently reduced seizures. No negative effect observed on longitudinal growth or bone mass after 24 months MAD treatment.*
Sixty-eight Dutch children on KDT for more than 6 months and who had at least two DXA scans [126]	Retrospective, observational cohort study	Changes in DXA-BMD in children with KDT and evaluation of efficacy of i.v. bisphosphonate therapy	*In 50% of patients, ≥ 1 LS-DXA scans performed. 8.8% got a fracture during KDT, and also 8.8% got kidney stones. Not significant BMD decrease. BMD significantly increased in the five patients treated with i.v. bisphosphonate therapy, vs. not treated.*
Three adult Italian patients with GLUT-1 deficiency syndrome on KD for > 5 years. Normocaloric KD on a 3:1 ketogenic ratio [129]	Case series report	Long-term effects of KD on body composition and bone mineral status in GLUT-1 deficiency syndrome; Anthropometric and body composition measurements (weight, height, BMI, waist circumference, abdominal circumference, skinfold-thickness measurement, arm muscle circumference, DXA-assessed body fat mass, lean body mass (LBM), BMC, and BMD.	*No appreciable changes in weight and body composition of adults with GLUT-1 deficiency. No evidence of potential adverse effects of KD on bone health*
Ten healthy, physically active UK men, age 24 ± 3 yrs., nonsmokers, not suffered a bone fracture or injury of any type in the previous 12 mo., free from musculoskeletal injury, not taking any medication, and not suffering from any condition known to affect bone metabolism. Two randomized, repeated-measures, counterbalanced 7-day experimental trials, involving either placebo (PBO) or CHO ingestion during 120 min of treadmill running [149]	Clinical trial	Immediate and short-term bone metabolic responses to CHO feeding during treadmill running; β-CTX, P1NP, OPG, OC, PTH, leptin, GLP-2, and IL-6, cortisol, insulin, serum calcium, albumin, and phosphate.	*CHO feeding during exercise attenuated the β-CTX and P1NP responses in the hours following exercise, indicating an acute effect of CHO feeding on bone turnover.*
Ten physically active UK men with at least one bout of endurance running per week, free from fracture in the previous 12 months and any condition known to affect bone metabolism. Free from musculoskeletal injury [149]	Clinical trial	Effect of an overnight fast vs. feeding of a single mixed meal, on the bone metabolic response to an acute bout of treadmill running; three-day food diary, β-CTX, P1NP, OC, OPG, cortisol, bone ALP, PTH, albumin-adjusted calcium, phosphate, leptin, and ghrelin	*Bone markers not significantly differ from baseline on follow up 1–4. Fasting had a minor effect on the bone metabolic response to subsequent acute, endurance exercise, reducing the duration of the increase in β-CTX during early recovery, but no effect on changes in bone formation markers. Reduction of duration of the β-CTX response with fasting not fully explained by changes in PTH, OPG, leptin or ghrelin*
Nine UK male runners, age 21 ± 1.9 years, completing a morning and afternoon high-intensity interval running protocol (interspersed by 3.5 h) under dietary conditions: high CHO availability, reduced CHO but high fat availability, or reduced CHO and reduced energy availability [150]	Clinical trial	Effects of post-exercise CHO and caloric restriction on the modulation of skeletal muscle cell signaling pathways as well as indicators of bone metabolism.	*Post-exercise circulating βCTX was significantly lower in high CHO compared to low CHO-high fat and low CHO and reduced energy availability*
Twenty-five males, and 5 females Australian World-class race walkers completing 3.5-weeks of energy-matched high CHO or ketogenic low-carbohydrate, high-fat diet followed by acute CHO restoration [152]	Clinical trial	Diet-exercise interactions related to bone markers in elite endurance athletes after a 3.5-week ketogenic low-CHO, high-fat diet and subsequent restoration of CHO feeding; serum CTX, P1NP, OC assessed at rest (fasting and 2 h post meal) and after exercise (0 and 3 h) at baseline, after the 3.5-week intervention (Adaptation) and after acute CHO feeding (Restoration)	*Markers of bone modeling/remodeling were impaired after short-term LCHF diet, and only a marker of resorption recovered after acute CHO restoration.*

Legend: KD = ketogenic diet; DXA = dual x-ray absorptiometry; PTH = parathyroid hormone; 25OHD = 25 OH vitamin D; BMI = body mass index; BMC = bone mineral content; BMD = bone mineral density; LS = lumbar spine; FN = femoral neck; GT = greater trochanter; CHO = carbohydrates; ALP = alkaline phosphatase; OC = osteocalcin; mo./mos. = month/months; βCTX = carboxy-terminal collagen crosslinks; P1NP = Procollagen I N—Terminal Propeptide; GLP-2 = Glucagon-like peptide-2; IGFBP3 = Insulin-like growth factor-binding protein 3; BSAP = bone specific alkaline phosphatase; uNTX = urinary N-terminal telopeptide collagen type 1, MAD = modified Atkins diet; SDS = standard deviations.

**Table 6 ijms-22-00435-t006:** General example of a lasting 24-months study protocol to evaluate bone health-related endpoints.

Clinical Parameters	Biochemical Parameters	Instrumental Parameters
Baseline Familial clinical history, physiological and pathological personal history (especially regarding clinical fracture risk factors); calcium/phosphate daily intake questionnaire; sun exposure questionnaire; drug history; presence/absence and type of physical activity; smoke and alcohol habitus. Tanner stage or bone age results should be applied in the pediatric age. Baseline, 6, 12, 18, and 24 mos. Height Weight BMI Waist circumference Clinical evaluation	Baseline, 6 mo., 12 mo., 18 mo., and 24 mo. Serum: 25OHD, PTH, calcium, phosphate, BTMs *°, total proteins, fractioned proteins, insulin, glucose, creatinine. Standard urine test and 24 hrs. urinary collection for: calcium, phosphate, and citrates To assess and confirm a stable ketosis state: early morning blood β–OHbutyrate and post-dinner urine acetoacetate. Alternatively, urinary stick tests for ketone bodies. * also, serum OPG, sclerostin, and IL-6 could be considered in the biochemical assessment. ° Follow-up BTMs testing should be done when its expected change equals or exceeds the least significant change (LSC) [157].	Baseline, 12 mo., 24 mo. DXA for LS, FN/Total hip, Nondominant BMC/BMD (when the population includes females prior to menopause and males younger than age 50 use Z-score; this is particularly important in children whose adjustments have to be made for growth, and interpretation) [163]. Follow-up BMD testing should be done when its expected change equals or exceeds the least significant change (LSC) [156]. One year after initiation or change of therapy is appropriate, with longer intervals once therapeutic effect is established [163] DXA Fracture Risk Assessment when subjects at high fracture risk are included [163] Baseline, 6, 12, 18, and 24 mos. Bio-impedancemetry

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
