# Peer review of "Energy Metabolism and Ketogenic Diets: What about the Skeletal Health? A Narrative Review and a Prospective Vision for Planning Clinical Trials on this Issue"

_ijms, 2021, doi:10.3390/ijms22010435_

Round 1

Reviewer 1 Report

This is a very thorough review on the role of ketogenic diet in bone regulation. The authors specifically discuss the biochemical basis of ketogenic diets in relation to cell and molecular processes in bone which determine its function. This is complemented by an overview of clinical studies that present parameters of bone health outcomes in different ketogenic diet regimes. This study presents a valuable tool for scientists and clinical professionals who are investigating this still relatively unexplored association between bone health and ketogenic metabolism. However the authors are kindly asked to pay attention to the following comments:

Abstract

  • “However, the causes of such negative impact on bone health has to be better defined, apart from the hypothesized effects of diet itself, generic alterations in vitamin  D and growth factors production and metabolism, other than the concomitant use of antiepileptic drugs and the impaired mobilization in a large proportion of these patients.”: this sentence is too long which complicates the structure.

Introduction

  • The biochemical pathway figure is not indicated as figure 1.
  • Figure 2 is appears cut. Please annotate the shown tissues.
  • In figures 1,2,3 the font does not have sufficient contrast.
  1. “…adipocytes share a common mesenchymal cell progenitor and that bone cell…”: you mean mesenchymal stem cells?
  2. “…In the early tetrapods, a vertebrate superclass with four limbs prevalently adapted to life in a subaerial environment, the evolution of…”: please change to In the early tetrapods, a vertebrate superclass with four limbs prevalently adapted to life in a subaerial environment; the evolution of

3.1 “It has been described that at the skeletal level insulin signaling stimulates…”: please change to It has been described that at the skeletal level of insulin signaling stimulates

3.1 “thus reducing calcification process even in the presence of strong”: please change to thus reducing the calcification process even in the presence of strong

- Please improve the format of table 1. In the current form it does not aid the reader.

- “to influence body composition”: please mention more specifically what is meant.

3.1 “It is also  important to consider that bone marrow stromal stem cells (BMSCs), when exposed to chronic high  glucose levels, show a peroxisome proliferator-activated receptor gamma (PPARγ)-dependent  activation enhanced adipogenic activity, rather than the osteogenic one, as also an increased  expression of cyclin D3 [26] together with a decreased Runx2 [27], ALP [28], and OC expression in 179 OBLs.”: please simplify this sentence.

3.1 and 3.2: 3.1 describes signaling pathways in different types of bone cells but these are introduced later in 3.2, please restructure these paragraphs to ensure a more logical order in introduction of concepts.

3.2 “Once their task is finished, they give rise to the bone lining cells or 201 osteocytes [33], acting as mechanical sensors [34] and secreting specific molecules, osteokines, such 202 as sclerostin that binds to its receptors on the cell surface of OBLs, ultimately inhibiting osteoblastic 203 differentiation and bone formation.”: please improve sentence structure.

3.3 “Specifically, BMP signaling has been demonstrated to correlate with several signaling pathways such as mammalian target of rapamycin  (mTOR), known to be activated by amino acids, glucose, insulin and other hormones regulating the  metabolism, hypoxia-inducible factor (HIF), Wnt, and self-degradative process autophagy in  coordination of both energy metabolism and bone homeostasis [43].”: it is not clear what is meant by correlation. Please simplify the sentence.

  1. “proliferative BMSCs and osteoprogenitors use”: are osteoprogenitors different than BMSC? Please clarify here as well as throughout the text.

- Similarly to Table 1, table 2 does not help the reader in the current form.

- HIF-1a is several times mentioned but there is little introduction about the role of hypoxia in bone biology or metabolism.

- Table 4 contains too much text, please state the information in a more concise manner to provide more of an overview.

5.3 It is not clear what is meant by acid-ash proteins.

- Table 5. Please try to reduce text where possible.

6.1 “i.e. stating that there is a variation when in reality there isn't)”: please change to i.e. stating that there is a variation when in reality there is not)

General

  • Inflammation is an important aspect of bone health which can also be regulated through diet, please elaborate on this.
  • An important point is the disruption of the reading flow by sentences which are too long, as a result of which concepts appear confusing. Please revise the text by introducing a clear structure, shorter paragraphs and more concise sentences.

Author Response

Rebuttal to Reviewer 1

The authors and, in particular, the undersigned corresponding author are very grateful to this reviewer for his/her overall judgment and for his/her absolutely correct and adequate comments that have helped to improve our paper. Below, all the answers/changes made, point by point, to the reviewer’s observations. The capital letter R indicates the comments/questions made by the reviewer and the capital letter A the answers/changes provided by us.

All the corrections/modifications are highlighted in yellow within the text.

R: Comments and Suggestions for Authors

This is a very thorough review on the role of ketogenic diet in bone regulation. The authors specifically discuss the biochemical basis of ketogenic diets in relation to cell and molecular processes in bone which determine its function. This is complemented by an overview of clinical studies that present parameters of bone health outcomes in different ketogenic diet regimes. This study presents a valuable tool for scientists and clinical professionals who are investigating this still relatively unexplored association between bone health and ketogenic metabolism. However, the authors are kindly asked to pay attention to the following comments:

R: Abstract

“However, the causes of such negative impact on bone health has to be better defined, apart from the hypothesized effects of diet itself, generic alterations in vitamin D and growth factors production and metabolism, other than the concomitant use of antiepileptic drugs and the impaired mobilization in a large proportion of these patients.”: this sentence is too long which complicates the structure.

A: the sentence has been now modified and shortened as requested. Highlighted in yellow, the new sentence is “In these subjects, the concomitant use of antiepileptic drugs and the reduced mobilization may partly explain the negative effects on bone health, but few is known about the effects of diet itself, and/or generic alterations in vitamin D and/or impaired production of growth factors.”

R: Introduction

The biochemical pathway figure is not indicated as figure 1.

A: Now, the biochemical pathway figure is indicated as Figure 1.

R: Figure 2 is appears cut. Please annotate the shown tissues.

A: Figure 2 is not cut and the shown tissues are now annotated.

R: In figures 1,2,3 the font does not have sufficient contrast.

A: The font has been now improved in all the figures

R: “…adipocytes share a common mesenchymal cell progenitor and that bone cell…”: you mean mesenchymal stem cells?

A: The reviewer’s observation is correct. The sentence now is “Bone cell progenitors, skeletal muscle cells and adipocytes share a common mesenchymal stem cell as progenitor…”

R: “…In the early tetrapods, a vertebrate superclass with four limbs prevalently adapted to life in a subaerial environment, the evolution of…”: please change to In the early tetrapods, a vertebrate superclass with four limbs prevalently adapted to life in a subaerial environment; the evolution of..

A: The sentence has been changed as suggested “In the early tetrapods, a vertebrate superclass with four limbs prevalently adapted to life in a subaerial environment; the evolution of...”

R: 3.2 “It has been described that at the skeletal level insulin signaling stimulates…”: please change to It has been described that at the skeletal level of insulin signaling stimulates

A: The sentence has been changes as suggested “At the skeletal level, insulin signaling stimulates either the OC expression or the OBLs differentiation, by inhibiting…”

R: 3.2 “thus reducing calcification process even in the presence of strong”: please change to thus reducing the calcification process even in the presence of strong

A: The sentence has been changed as suggested “thus reducing the calcification process even in the presence of strong osteogenic stimulants.”

R: - Please improve the format of table 1. In the current form it does not aid the reader.

A: The format of Table 1 has been improved for a simplified reading

R: - “to influence body composition”: please mention more specifically what is meant.

A: The sentence has been deeply modified and now is “a significant association between BGLAP (OC gene) genetic variants and BMI in healthy subjects has been reported to be most likely associated with body mass as composite phenotype and less likely associated with adipose tissue itself, even though it cannot be excluded that not only the BGLAP gene may cause the association observed.”

R: 3.2 “It is also important to consider that bone marrow stromal stem cells (BMSCs), when exposed to chronic high glucose levels, show a peroxisome proliferator-activated receptor gamma (PPARγ)-dependent activation enhanced adipogenic activity, rather than the osteogenic one, as also an increased expression of cyclin D3 [26] together with a decreased Runx2 [27], ALP [28], and OC expression in 179 OBLs.”: please simplify this sentence.

A: This sentence has been now simplified “Interestingly, the bone marrow stromal stem cells (BMSCs) exposed to chronic high glucose levels show an enhanced adipogenic activity by peroxisome proliferator-activated receptor gamma (PPARγ)-dependent mechanism, as also an increased expression of cyclin D3 [34], with a decreased Runx2 [35], ALP [36], and OC expression in OBLs.”

R: 3.1 and 3.2: 3.1 describes signaling pathways in different types of bone cells but these are introduced later in 3.2, please restructure these paragraphs to ensure a more logical order in introduction of concepts.

A: Thanks to reviewer. According to the right reviewer’s suggestion these paragraphs have been definitively restructured following a more logical order.

R: 3.1 “Once their task is finished, they give rise to the bone lining cells or 201 osteocytes [33], acting as mechanical sensors [34] and secreting specific molecules, osteokines, such 202 as sclerostin that binds to its receptors on the cell surface of OBLs, ultimately inhibiting osteoblastic 203 differentiation and bone formation.”: please improve sentence structure.

A: We fully agree with this suggestion. Now the structure of the sentence has been improved “Once completed bone formation, OBLs give rise to Ocs [21] acting as mechanical sensors [22] and secreting specific osteokines, such as sclerostin. Sclerostin binds to the low-density lipoprotein receptor-related protein 5 and 6 (LRP5/6) receptors on OBLs cell surface and inhibits the Wnt signaling pathway determining anti-anabolic effects on bone formation [23].”

R: 3.3 “Specifically, BMP signaling has been demonstrated to correlate with several signaling pathways such as mammalian target of rapamycin (mTOR), known to be activated by amino acids, glucose, insulin and other hormones regulating the metabolism, hypoxia-inducible factor (HIF), Wnt, and self-degradative process autophagy in coordination of both energy metabolism and bone homeostasis [43].”: it is not clear what is meant by correlation. Please simplify the sentence.

A: The sentence has been now simplified “Specifically, BMP signaling demonstrated to interact with several signaling pathways such as the mammalian target of rapamycin (mTOR), activated by amino acids, glucose, insulin, hypoxia-inducible factor (HIF), Wnt, and self-degradative process autophagy to coordinate both energy metabolism and bone homeostasis [44].”

R: “proliferative BMSCs and osteoprogenitors use”: are osteoprogenitors different than BMSC? Please clarify here as well as throughout the text.

A: The text has been now clarified “In fact, proliferative BMSCs and cells already at an osteogenic differentiation stage, use essentially glycolysis as…”

R: - Similarly to Table 1, table 2 does not help the reader in the current form.

A: The format of Table 2 has been improved for a simplified reading

R: - HIF-1a is several times mentioned but there is little introduction about the role of hypoxia in bone biology or metabolism.

A: A more in-depth introduction about hypoxia and bone biology has been now made “Interestingly, either pathological or environmental hypoxic conditions appear to influence bone health, since bone cells exhibit to be distinctly responsive to hypoxic stimuli, even if they act in a negative or positive way is still unknown. It has been suggested that hypoxia may induce an osteogenic-angiogenic response, but also stimulate excessive OCL activity. Moreover, several hypoxia-associated factors may influence bone metabolism by determining changes in energy metabolism and increasing the generation of reactive oxygen species (ROS) as also impair the physiological acid–base balance [58].”

R: - Table 4 contains too much text, please state the information in a more concise manner to provide more of an overview.

A: Information in Table 4 is definitively more concise with reduction of the contained text.

R: 5.3 It is not clear what is meant by acid-ash proteins.

A: exemplification of acid-ash concept is now reported in the brackets “(acid‐ash hypothesis identifies protein and grain foods as cause of the calcium release from the skeleton to buffer the acid load from the diet and increase urinary excretion of calcium)” and a specific reference has been introduced both in text and bibliography “[155]”.

R: - Table 5. Please try to reduce text where possible.

A: The text of Table 5 has been significantly reduced (one page less)

R: 6.1 “i.e. stating that there is a variation when in reality there isn't)”: please change to i.e. stating that there is a variation when in reality there is not)

A: We thank you again the reviewer. The sentence has been changed into “(i. e. stating that there is a variation when in reality there is not)”

R: General

Inflammation is an important aspect of bone health which can also be regulated through diet, please elaborate on this.

A: In perfect agreement with the reviewer's suggestion, paragraph 7, entitled “Inflammation, diet and bone health”, was introduced, trying to keep us within a text dimension that is as concise as possible.

R: An important point is the disruption of the reading flow by sentences which are too long, as a result of which concepts appear confusing. Please revise the text by introducing a clear structure, shorter paragraphs and more concise sentences.

A: The text has been thoroughly revised with introduction of a clearer structure. Paragraphs have been shortened and sentences are more concise now (please, see all the yellow highlighted sections throughout the paper).

Reviewer 2 Report

The information is vague and sources are random, manuscript has to be significantly improved in order to get published.

Extensive edits for grammar and sentence cohesion are necessary, e.g. first word of the abstract is not even capitalized.

Figures should be created by the authors. Please provide high resolution pictures.

Author Response

Rebuttal to Reviewer 2

The authors are very grateful to this reviewer for his/her overall judgment and for his/her clarification that have helped to improve our paper. Below, all the answers/changes made to the reviewer’s observations. The capital letter R indicates the comments/questions made by the reviewer and the capital letter A the answers/changes provided by us.

Overall text, including main text, Figures and Tables, has been thoroughly revised. We introduced a clearer structure. Paragraphs have been shortened and sentences are more concise now (please, see all the yellow highlighted sections throughout the paper). All the corrections/modifications are highlighted in yellow.

R: Comments and Suggestions for Authors

The information is vague and sources are random, manuscript has to be significantly improved in order to get published.

A: We fully agree with what this reviewer has expressed. The main problem is that it is this specific topic that is "vague" and “poorly documented” in the literature which makes it difficult to disclose. As also reported in the text, it is necessary that clinical studies, aiming to understand whether a correlation between ketogenic diets and bone health exists, are properly planned and designed. There are too many studies on small samples, with bias in the selection of the subjects included, or of inadequate duration for the intended clinical outcome. There is a wide discretion, not justified by culturally appropriate bases, in the choice of skeletal, clinical, biochemical and/or instrumental parameters, used with the aim of evaluating an impact, whether positive or negative, of these diets on biology and metabolism of human bone tissue. One of the goals that we, as bone specialists, set ourselves is to ensure a multidisciplinary vision to this topic. Therefore, we have introduced paragraphs and tables concerning the major pitfalls of this issue, with useful suggestions for planning an adequate experimental design of clinical trials in this field.

R: Extensive edits for grammar and sentence cohesion are necessary, e.g. first word of the abstract is not even capitalized.

A: An extensive edit for grammar and sentence cohesion has been made by a native English speaker and the first word of the abstract, an obvious mistyping error, is now capitalized.

R: Figures should be created by the authors. Please provide high resolution pictures.

A: The figures were assembled in order to “create” new ones, not already present in the literature or online. Now the pictures resolution is higher.

Round 2

Reviewer 1 Report

The article has been slighly improved but not enough for publication, especially to with regard to the following points: 

  1. Language remains in several places quite awkward which makes concepts a bit ambiguous for the reader.
  2. Tables do not provide aid the commucation of the important points, in the current form they are more a list of bullet point statements. 
  3. Structure of the text still needs to be improved in order to introduce concepts in a logical order and a concise manner. 

Examples of language errors: 

Abstract

  • Please change: “effects on bone health, but few is known about the”: to “effects on bone health, but little is known about the”

3.1

- Please change “throughout the life” to “throughout life”.

Table

  • Please change “Summarization of the main”: to “Summary of the main”

3.3

- Please change “Specifically, BMP signaling demonstrated to interact with” to “Specifically, BMP signaling was demonstrated to interact with”

Author Response

To Reviewer

First of all, these authors are deeply grateful to the reviewer for the hard work done in reading our review and for the efforts profuse to provide suggestions and comments aimed at improving the final product. We understand well, and appreciate very much, the patience that the reviewer had to read a paper that aims to give a body to a topic which, in fact, is like an "anomalous" puzzle by its nature made up of pieces that do not perfectly match one another. In fact, we authors ourselves too have faced this hard task of giving a sense and a logical order to the interlocking of these "imperfect" tiles, through a difficult and complex reading of many works, each with its own relative importance, which, in fact, have not tried to provide a more holistic version of the argument.

However, it is equally true that, as bone specialists involved for many years in the study of the pathophysiological processes underlying metabolism and skeletal health, it is time to start giving a dignity of its own to this issue, suggesting the establishment of a sub-chapter culturally identifiable as "osteodietology". We do not know, with great humility, if we will really have succeeded in this task, but we had the moral obligation to try and, therefore, we hope that this "complex" review can only represent the beginning of a new scientific path that will receive many others input, much better in terms of ways and contents than our contribution can do. In particular, we believe to be very important that the vision of prevention and treatment of bone fragility can no longer be limited to an exclusively pharmacological vision or generic advice on carrying out "adequate" physical activity and calcium and vitamin supplementation D.

Thanks again to our reviewer!

Any revisions has been clearly highlighted, and the "Track Changes" function in Microsoft Word has been used to make them easily visible to the editors and reviewers.

The capital letter R indicates the reviewer’s comment/answer.

The capital letter A indicates the authors’ comment/answer.

Specific points

R: "Comments and Suggestions for Authors
"

The article has been slightly improved but not enough for publication, especially to with regard to the following points:

Language remains in several places quite awkward which makes concepts a bit ambiguous for the reader.

Tables do not provide aid the communication of the important points, in the current form they are more a list of bullet point statements.

Structure of the text still needs to be improved in order to introduce concepts in a logical order and a concise manner.

A: As suggested by reviewer, always together with an English mother tongue consultant, we thoroughly reviewed the whole paper, hoping to have made it less " quite awkward" in the points defined as such. The points of possible ambiguity have been modified to give them a clearer and more intelligible form.

R: Examples of language errors:

Abstract

Please change: “effects on bone health, but few is known about the”: to “effects on bone health, but little is known about the”

A: the sentence has been change as requested

R: 3.1

- Please change “throughout the life” to “throughout life”.

A: This change has been now made

R: Table

Please change “Summarization of the main”: to “Summary of the main”

A: At Table 1 now is written “Summary of the main…”

R: 3.3

- Please change “Specifically, BMP signaling demonstrated to interact with” to “Specifically, BMP signaling was demonstrated to interact with”

A: the text has been changed into “Specifically, BMP signaling was demonstrated to interact with…”

R: Tables do not provide aid the communication of the important points, in the current form they are more a list of bullet point statements.

A: we have now modified the best we could the Tables content, but we honestly believe that in the specific context of our review the tables are of help to synthesize some knowledge that, otherwise, we would have to insert in more detail in the text, further lengthening it and making it smooth in its already difficult reading.

R: Structure of the text still needs to be improved in order to introduce concepts in a logical order and a concise manner.

A: In accordance with what rightly suggested by the reviewer, we have modified the structure of the text, hoping to have significantly improved it, and given a more logical order to the arguments. In particular, we have given a logical, spatial sequence to the paragraphs concerning the description of the various roles, tasks and hypotheses related to the Wnt pathwys (see also the numbering provided) and, where possible, we have introduced new sub-paragraphs, both to reduce the heaviness of some previous paragraphs, and because we hope to have improved and facilitated understanding.

Round 3

Reviewer 1 Report

The level of English has been greatly improved in this version, authors are still advised to check one more time throughout the text for small errors (example table 4 "cortical and trabecular bone micro-CT" should be "Cortical and trabecular bone micro-CT". 

Also tables 1-3 still seem to be more a summary of facts (more like a text box) rather than tables. 

Author Response

To Reviewer

These authors continue to sincerely thank the commitment and advice lavished by this reviewer who has greatly contributed to improving, as he himself has attested, the quality of this paper.

Any revision made has been clearly highlighted, and the "Track Changes" function in Microsoft Word has been used to make them easily visible to the editors and reviewer.

The capital letter R indicates the reviewer’s comment/answer.

The capital letter A indicates the authors’ comment/answer.

Specific points

A: The level of English has been greatly improved in this version, authors are still advised to check one more time throughout the text for small errors (example table 4 "cortical and trabecular bone micro-CT" should be "Cortical and trabecular bone micro-CT".

R: The text has been carefully checked throughout the text for small errors and, consequently, correct (see the Track Changes). Now the sentence “Cortical and trabecular bone micro-CT” has been reported in table 4.

A: Also, tables 1-3 still seem to be more a summary of facts (more like a text box) rather than tables.

R: The tables 1-3 have been modified to make them more in the form of real tables, organized in columns, each with its own title. We hope, with the utmost sincerity, to have succeeded in our intent.

Finally, the authors sincerely appreciated the work done by the reviewer, who, with a true spirit of service, performed his/her role superbly. This for us is a good example of the real importance of what it means to be of a good reviewer.